



# Surface movement above an underground coal longwall mine after closure

**André Vervoort** *(corresponding author)*

Department of Civil Engineering, KU Leuven, Leuven, Belgium [Kasteelpark Arenberg 40, 3001
Leuven, Belgium]
*Correspondence to: A. Vervoort (andre.vervoort@kuleuven.be)*

**Abstract.** The surface movement in an area of about 22 km² above the underground coal mine of
Houthalen was analyzed based on Interferometry with Synthetic Aperture Radar (INSAR)
measurements. After its closure in 1992, a residual subsidence was observed over a period of
several years, followed by an uplift of the surface above and around the past longwall panels,
whereby the rate of movement was, in absolute terms, of the same order for the two types of
movements. The processes behind these movements are different. The process of subsidence is
caused by the caving of the roof above the mined out area and is mainly a mechanical stress-
deformation process, including time-dependent aspects. However, the process of uplift is most
probably caused by the swelling of the clay minerals in the rocks after the flooding of the
underground workings. Hence, the areas in which there is the greatest risk of damage to the
surface infrastructure are not the same for the hazards linked to subsidence and uplift. For
example, the zone in which the maximum uplift occurs clearly is at a different location from that
of the zone with the maximum residual subsidence. There is no clear sign that the amount of
mining underneath affects the residual subsidence, and there is no indication that the process of
uplift is linked directly to the mining characteristics. It is more likely that uplift as the result of
flooding is initiated at, or close to, the vertical shafts.

**Keywords:** coal mining; surface movement; subsidence; uplift; radar-interferometry

## 1 Introduction

Most research of the movements of the Earth's surface above underground mines has focused on
the direct impact of mining, i.e., the impacts that occur during the lifetime of the mine. This is
entirely logical because the largest amount of movement occurs during that period. Also, during
that period, the mining company can limit the hazards, e.g., by selecting a different mining
method (e.g., room and pillar instead of longwall), by backfilling the mined-out area instead of
creating a goaf, or by changing the mining geometry. However, by introducing the concepts of
sustainable mining, the long-term impact of mining on its surroundings has been receiving
greater attention. This means that the period after the mine's closure is a period that should not
be neglected. Surface movements after closure, which is the topic of this study, should be
investigated in detail. In the past decades, not only individual mines in Western Europe have





been closed, but coal production stopped in entire coal basins. As a consequence, the deep
underground was flooded because access to the underground facilities was sealed off, and the
underground pumping stations were dismantled. This created a new hazard, i.e., the uplift of the
surface caused mainly by the swelling of clay minerals (Bekendam and Pöttgens, 1995).
Although the order of magnitude of the movements in such uplifts is smaller than the subsidence
that occurs during mining, cases have been reported in which uplifts have damaged buildings and
the surface infrastructure (Baglikow, 2011; de Vent and Roest, 2013; Caro Cuenca et al., 2013).
So, studying this phenomenon is more than a pure scientific exercise. To date, other researchers
have focused mainly on understanding the phenomenon (e.g., Herrero et al., 2012) and
identifying general trends, whereby the link with the rise in water level was an important issue
(Caro Cuenca et al., 2013; Devleeschouwer et al., 2008). In this study, we tried to provide better
quantification of the movement after closure and the difference between the residual downward
movement and the ultimate uplift of the surface. To accomplish this, we studied the past mining
directly underneath the observation points.
The underground coal mine of Houthalen, Belgium, was closed in 1992. For a period of nearly
two decades (from 1992 through 2010), we analyzed the movements of the surface above the
mine based on radar-interferometry or Interferometry with Synthetic Aperture Radar (INSAR)
measurements. The production of coal in this mine began in 1939, and, in 1964, the mine was
merged (and connected underground) with the Zolder coal mine, which is situated to the west of
the Houthalen mine. Production was stopped in both mines in 1992, and the access was sealed
off. Hence, the underground pumps also were stopped, causing flooding of the underground
work areas, the surrounding rock mass and caved zones.
Longwall mining with goaf was the method used in the mines, and different coal seams were
mined. The area in which the detailed study of surface movement was conducted is situated from
Latitude 51.01°N to 51.05°N and from Longitude 5.33°E to 5.40°E, an area of approximately 5.0
(EW) by 4.4 (NS) km² (Fig. 1). At a certain X-Y position within the mined area, one to eight
different coals seams were mined. The combined mining height of the several seams varied from
2.0 to 12.3 m within this area. The height of the mining of individual panels varied from 0.9 to
2.7 m, and, normally, about 10 to 40 cm of it were layers of waste rock. In some cases, either no
waste rock was mined or only a few centimeters were mined, but, in other cases, almost 1 m of
waste rock was recorded as having been mined. As the map indicates, certain zones were not
mined. Apart from the zone around the vertical shafts (around the coordinates of Latitude
51.025°N and Longitude 5.370°E), these unmined zones mainly were areas around faults. The
latter were composed of a predominant set of NNW - SSE striking normal faults with
subordinate N-S to NE-SW striking thrust faults. In the later decades of production, a typical
longwall panel had dimensions of 200 by 800 to 1,000 m. The main and tail gates were
immediately adjacent to the panel, and they were just single tunnels with a horseshoe cross-
section. So, no barrier or remnant pillars existed between the longwall panels. In the area that we
studied, the mining depth varied from 539 to 967 m, and the mining occurred between 1932 and
1992. However, most of the panels were mined in the 1960s and 1970s. In Sect. 4, more details





of the mining characteristics are provided, and their possible influences on the surface
movements are discussed.
The coal strata in the Campine basin in northeast Belgium belong to the Upper Carboniferous
strata (Westphalian unit), the time of the formation of many coals fields in Europe (Langenaeker,
2000; Vandenberghe et al., 2014). The top of the Upper Carboniferous strata generally occurs at
depths of approximately 400 to 600 m. The waste rock within these coal strata is composed
mainly of shale, siltstone, sandstone, and thin (unmined) coal layers. The sandstone is classed as
medium-strong, with a typical Uniaxial Compressive Strength (UCS) of 90 MPa (Caers et al.,
1997). However, values up to 160 MPa also have been measured. The other types of rocks are
classified as weak rock, e.g., siltstone was tested with an UCS-value from 17 to 68 MPa with an
average of 46 MPa, and coal with an UCS-value from 6 to 10 MPa with an average of 7 MPa.
The average values of Young's modulus for these three types of rocks were determined as 28
GPa for sandstone, 9 GPa for siltstone, and 1 GPa for coal (Caers et al., 1997). Overall, the
successive strata are relatively thin (on the order of dm to m in scale). The overburden is
composed of weak to very weak geological material (e.g., sand, clay, and chalk). Several
aquifers and aquitards are present over the entire section of the overburden.
**2 Radar-Interferometry data**
Radar-interferometry or Interferometry with Synthetic Aperture Radar (INSAR) is a recent
remote sensing technique that allows the study of large time series of surface movements (Akcin
et al., 2010; Herrera et al., 2009; Hongdong et al., 2011; Jung et al., 2007; Zhenguo et al., 2013).
The movement of reflective surfaces (i.e., the so-called permanent scatterers) is followed during
successive cycles of the satellite. There is high spatial coverage of the areas studied, at least if
the area corresponds to a built environment. In comparison to conventional leveling methods, the
advantages of radar-interferometry include (i) large areas can be covered for the same effort
(e.g., a full concession area of a mine), (ii) measurements are conducted on a regular and
frequent basis (i.e., one measurement per satellite revolution (35 days for the datasets used in this
research)), and (iii) a dense network of reflectors is available (sometimes every 10 to 20 m). One
of the disadvantages is that, when no reflective surfaces are identified in a specific zone, no
information is available on the movement of the surface. For example, this was the case for the
area studied in the zones composed of agricultural land, woodland and unused or semi-natural
land. Other problems were that 1) the recorded movement corresponds to the reflection of a
surface area of 4 x 20 m and not of a discrete point and 2) that the recordings are not of the
Earth's surface but of reflective objects, which can be hardened surfaces, such as the roofs of
buildings (for the most part), as well as parking lots and roads. This means that for buildings the
type and depth of the foundation and the structure itself affect the movement of the reflector
(Dang et al., 2014).



In this study, the European C-band ERS1/2 and ENVISAT-ASAR satellite images were used,
which were available for research through a European Space Agency (ESA) research proposal
(Devleeschouwer et al., 2008). The recorded periods were for both sets from August 1992
through December 2000 (87 cycles of 35 days) and from December 2003 through October 2010
(72 cycles of 35 days), respectively. Generally, it is accepted that the linear velocities can be
estimated with accuracies of about 1 mm per year (Marinkovic et al., 2009; Sousa et al., 2009).
However, these values depend significantly on the number of images and the conditions in which
they were obtained as regards baselines, Doppler centroid distribution, selected pixel density,
how they are connected, and the presence of atmospheric effects.
**3 Analysis of surface movement**
Earlier research (Vervoort and Declercq, 2016) showed that, in this area at the end of the first
period of observation (from August 1992 through December 2000), uplift had already been
initiated in certain zones or for certain reflectors. In a similar way, it was observed that, at the
start of the second period of observation (from December 2003 through October 2010), certain
reflectors were still undergoing downward movement. Therefore, the focus was placed on the
first five years of the first observation period (from mid-August 1992 through mid-August 1997)
and on the last 5 years of the second observation period (mid-September 2005 through mid-
September 2010). For comparison purposes, the last 2.5 years of the first period and the first 2.5
years of the second period also were analyzed partially. These two 2.5-year time zones were
from July 1998 through December 2000 and from December 2003 through June 2006,
respectively. This means that there was a gap between the time zones of 5 and 2.5 years in the
first period and a small overlap in the second period, but the main advantage of doing so was that
all time zones could be compared more easily. Hence, all scales for the graphs that correspond to
the 2.5-year time zones are halved.
In this research, downward movement has a negative sign, and uplift has a positive sign; the
same convention was used for the rate of movement (e.g., per year). However, when discussing
the smallest (minimum) movement or the largest (maximum) movement, we considered the
absolute value of the movement; in other words, when discussing the minimum rate, we did not
apply the pure mathematical definition of minimum. For the area studied, no public data were
available for the subsidence that occurred prior to the satellite observations.
**3.1 First observation period, characterized, on average, by subsidence**
In the five years from mid-August 1992 through mid-August 1997, the area studied was
characterized by an overall downward movement (Table 1 and Fig. 2a). Only two out of 1,073
reflectors were characterized by small upward movements, i.e., 3 and 6 mm. In the overall
picture, these can be neglected. Among the reflectors, 69% underwent residual subsidence





ranging from -20 to -40 mm over the five-year time zone (Fig. 2a). The average subsidence was -
33 mm, corresponding to an annual subsidence rate of about -6 mm. The maximum rate for the
studied area was about -16 mm per year (or a total of -80 mm). The distribution was slightly
positive, i.e., a long tail for the larger subsidence movement. Also the spread (e.g., maximum
minus minimum) was relatively large, i.e., 85 mm. Earlier research showed that the variation was
even larger if one considers the annual increases, i.e., the subsidence for each individual year,
and not the total subsidence divided by five years (Vervoort and Declercq, 2016). For the time
zone considered, the maximum increase per individual year was about -33 mm, i.e., five times
the average rate over the five years (-6 mm/year)  and about twice the maximum average rate
over the five  years (-16 mm/year).
If one looks at the spatial variation of the total surface movement over the five-year time zone, it
is apparent that the largest residual subsidence occurred in the northern Central part of the area
studied (Fig. 3a). Unfortunately, the reflectors were not evenly spread over the entire area. There
were several zones with dimensions of a kilometer wide in which there were no reflectors at all.
These zones, in this particular case were farm land, woodland, unused land, and a lake. The
combination of large zones without reflectors and the large fluctuation between neighboring
points motivated us to present the individual reflectors instead of calculating a contour map. The
latter would result in too much loss of detail and would result in large uncertainties for certain
sub-zones.
Fig. 3a shows that, although large movements may occur next to small movements, clustering is
present. For example in the western and southeastern parts of the area studied, the reflectors were
characterized mainly by a residual subsidence of -20 to -30 mm over the five years. Most
reflectors with a total residual subsidence over the five years of -60 mm or more correspond well
with the mined out area underneath (Fig. 1). In Sect. 4.2, a more detailed analysis is presented
with the past exploitation. To better visualize the map of reflectors, a north-south section is
presented in Fig. 4a. Ideally, one should present a very narrow zone in the east-west direction.
However, a compromise had to be found between a sufficient amount of reflectors over the entire
north-south length and not including too much variation in the east-west direction. Therefore, a
north-south zone was selected for Longitude between 5.37°E and 5.38°E, about 700 m wide. A
slightly longer transect was chosen, as shown in Figs. 1 and 3. No exploitation took place more
to the north than Latitude 51.05°N in the transect selected or its immediate surroundings; the
same is true more to the south than Latitude 51.005°N (Van Tongeren and Dreesen, 2004). In the
northern and southern part the residual movement was still a subsidence but the values were
small. In the zone between Latitude of 51.015°N and 51.050°N, mainly movements of -20 mm
and more were observed, with the largest values situated between 51.035°N and 51.045°N. The
variation of the observed values was about 20 mm. This can be explained by the inaccuracy of
the method, by the variation in the east-west direction, and by the local variation between
neighboring points.
For the same (first) observation period, the last 2.5 years also were analyzed (Table 1 and Fig.
2b). As mentioned earlier, all scales were halved to make the comparison easier. About 8.5% of



the reflectors underwent uplifts during this time zone of 2.5 years (Fig. 2b). Fig. 3b presents the
locations of the corresponding reflectors. It is very clear that these locations are complementary
to the zone of the largest residual subsidence observed in the first five years (Fig. 3a). The
maximum subsidence rate observed was about the same as during the first five years, i.e., about -
16 mm/year. The average rate was much smaller, i.e., -3.6 mm/year instead of -6.5 mm/year.
When looking at the north-south transect (Fig. 4b), large subsidence values occurred in a similar
area as in the five-year time zone. A peak was observed at a Latitude of about 51.04°N. The
variation of the observed values remained about 20 mm. As illustrated above, there was a non-
negligible number of reflectors characterized by uplifts, also above the unmined areas.
**3.2 Second observation period, characterized, on average, by an uplift**
In the five-year time zone from mid-September 2005 through mid-September 2010 at the end of
the second observation period, it was clear that the surface of the entire area had moved upwards
(Fig. 5a). Only six of the 1,808 reflectors had a slight downwards movement over this time
period. The smallest movement was -10 mm (Table 2). About 75% of the reflectors underwent
uplifts that ranged from 30 to 60 mm. The largest movement of all reflectors was 84 mm,
corresponding to an average rate of about 17 mm per year, while the global average movement
was 44 mm or nearly 9 mm per year. This means that the rate of uplift was larger than the
residual subsidence rate in the five-year time zone following the closure of the coal mine (e.g.,
an average rate of -6 mm/year vs. 9 mm/year). The shape of the distribution was negatively
skewed, which means that only a few reflectors were observed with small values. As for the first
five years in period 1, the variation in the increase of the uplift per individual year was larger
than the average rate over the five-year time zone. The maximum individual annual increase was
31 mm.
There was a clear difference between the start and end of the second observation period. Fig. 5b
shows the distribution of the increase in surface movement over the 2.5 years between December
2003 and June 2006. About 6% of the reflectors still had undergone a subsidence (in comparison
to 0.3% in the last five years). The classes between 5 and 20 mm of total increase (corresponding
to an average annual rate between 2 and 8 mm per year) contained about 71% of the reflectors
for the first 2.5 years, while the classes for the same annual rate (i.e., from 10 to 40 mm total
increase) was only about one third at the end of the second observation period.
The map of the reflectors in the area studied now shows a completely different picture (Fig. 6a)
in comparison to the first observation period. The largest uplift values were observed mainly in
the central to southern part. In the northern part, where the largest residual subsidence was
recorded, small uplift values were observed. In the 2.5-year time zone, from December 2003
through June 2006, numerous reflectors still had undergone subsidence in that northern part (Fig.
6b). In the eastern part (longitude larger than 5.39°E) of the southern half, subsidence was still
recorded, while this part was characterized by relatively small residual subsidence in the first
observation period. (Compare Fig. 3a with Fig. 6b.)



Fig. 7 presents north-south transects that are similar to those for the first observation period. In
the last five years of observation (Fig. 7a), the maximum uplift was observed at a Latitude of
about 51.024°N. Less than 10 mm of uplift were recorded farther to the south than 50.994°N and
farther to the north than 51.050°N. These zones were not situated above exploitation panels;
however, it is still worthwhile to note that there were uplifts in these areas. As for the subsidence,
a variation of about 20 mm for a specific coordinate was observed again. Overall the curve is
relatively symmetric. For the first 2.5 years of the second observation period (Fig. 7b), the values
along this north-south transect confirm what was mentioned before, i.e., most downward
movement was situated in the northern and southern parts, while the peak in uplift became
visible somewhere between 51.02°N and 51.03°N.
In comparison to these north-south transects, the east-west transects had a smaller variation,
except, of course, that the movement evolved to zero away from the exploitation in the east. In
comparison to the east of the area studied, where there was no exploitation, the exploitation of
the Zolder mine bordered the exploitation of the Houthalen mine in the west. So, this clearly
affected the movement. As an example, an east-west transect is presented in Fig. 8 for a Latitude
between 51.018°N and 51.026°N, which corresponds to the maximum uplift in the north-south
transect. More to the east than a Longitude of 5.41°E, no reflectors were available as there is
over a distance of about 3.5 km a nature reserve (without buildings or infrastructure). Although
the variation is small between a Longitude of 5.33°E and 5.39°E, the earlier chosen north-south
transect (5.37-5.38°E) was at or close to the east-west maximum.
**4 Discussion of results**
**4.1 Location of maximum residual subsidence vs. location of maximum uplift**
As mentioned above, the movement is more complex than can be represented by a single value.
Hence, one should be careful in replacing the individual measured values by an average or by a
smoothed curve. However, for comparison purposes, such smoothed curves were drawn for the
north-south transects, presented above. For the smooth curves of both of the five-year time zones
that were studied, the following observations were made (Fig. 9a):
- The absolute movement over five years is the same order of magnitude as the residual
subsidence and the uplift.
- The maximum observed movements were at different locations. In the phase of residual
subsidence, the maximum was situated around a Latitude of 51.04°N, while for the uplift phase,
the maximum was observed around 51.020°N to 51.025°N. This is in agreement with the maps
in Figs. 3a and 6a.
- To the northern and southern end of the chosen transect, the movements evolved towards zero,
away from the exploitation.


- The curve of the uplift is very symmetric, which is not the case for the curve of the residual
subsidence. In Sect. 4.3, the mechanism behind the uplift is further discussed.
All these points are also visible when looking at the smoothed curves for both 2.5-year time
zones that were studied (Fig. 9b). The only difference is that, in the southern part (more to the
south than a Latitude of 51°N), on average, there already was an uplift in the first observation
period, while a subsidence was observed, on average, in the second observation period.
When looking in detail at the movements at the locations of both maxima, the above can be more
quantified (Table 3 and Figs. 10 and 11). Around both maxima, the 10 nearest reflectors were
selected. The reflectors were not necessarily the same for the two observation periods, but they
were the same within each of the two observation periods. The amount of 10 reflectors is a
compromise between zooming in on a particular area and having enough data to be statistically
representative. Table 3 presents the minimum, maximum, and average of the total vertical
movement over the five-year time zone. The variation of these values as a function of time is
plotted in Figs. 10 and 11. As could be expected based on Fig. 9a, the difference between the two
groups of curves is clear. For the first observation period, there was a small overlap between the
two groups, i.e., the minimum of the residual subsidence of the location of the maximum residual
subsidence was slightly smaller than the maximum of the other location studied, but the
difference between the two averages was 27 mm over the five-year time zone. For the second
observation period, there was no overlap between the two groups. The difference between their
averages over the five years was 20 mm.
Table 3 provides a summary of some basic information on the exploitation just underneath the
two locations. Fig. 12 also indicates these locations. Under the maximum of the residual
subsidence, the mining was more recent than under the maximum of the uplift. Mining took
place in the periods of 1968-1982 and 1939-1959, respectively. However, 1982 was still 10 years
before closure (and the start of observation). A corner of a panel, which was mined in 1992 at a
depth of 820 m, is situated at about 250 m to the west of the location Max $_{RES\ SUBS}$. This means
that this location is within the zone of influence of that panel. However, on the E-W transect
(across the panel), we did not observe any maximum in residual subsidence above the most
recent panel. When comparing the mining depth, mining height, and the number of panels mined
underneath the two locations, the mining characteristics were rather similar. So, this means that,
apart from possibly the time of mining, there was no clear indication concerning the causes of
the difference between the movements of the two locations. In the next two paragraphs, more
locations are compared, which will indicate whether the effect of the time of mining is
significant.
**4.2 Influence of mining characteristics on residual subsidence after closure**
Mining by the longwall method results in caving above the mined-out areas, creating the goaf
area. A roof height of two to eight times the mined height generally is considered to be sufficient
to fill up the mined height, plus the caved height (Peng, 1986). In the Campine basin, an average



value of five times normally was assumed, corresponding to a bulking factor of 1.2. The caved
zone is composed of blocks of broken material and includes a large amount of small and large
cavities. Hence, Young's modulus of this caved material is several orders of magnitude smaller
than that of the original intact layers (Galvin, 2016). Over time, this volume is compressed
progressively, but it will never reach its original state. Apart from the caving of the immediate
roof layers, the rock further away fractures and sliding along the induced fractures occurs. Still
further away from the mining depth (i.e., closer to the surface), plastic and elastic deflections of
layers also occur. All these phenomena result in the occurrence of subsidence at the surface. A
typical trough shape is created, e.g., above and around a single panel that has been mined. The
zone of influence at the surface is larger than the dimensions of the panel itself. By considering
an angle of draw of 45°, as was often done for the Campine coal basin, the width of the zone of
influence is about the depth of mining, which varied from 539 to 967 m in the area studied. For
the coal basin that we studied, typical subsidence rates were 30 to 60 mm/month in the months
following the mining. Unfortunately, for the area studied, no public data were available for the
subsidence that occurred prior to the satellite observations. Worldwide, the maximum subsidence
ranges from 40 to 90% of the total mining height (Wagner and Schümann, 1991; Sheorey et al.,
2000). In the Campine basin, values of 80 to 90% normally are used. This means that, for the
area studied with a mining height varying from 2 to 12.3 m, a subsidence of 1.6 to 11.1 m could
have occurred. There is no reason to assume that the general rules of the amount of residual
subsidence following years or decades after mining would be any different from what can be
considered as the globally accepted knowledge, e.g., more subsidence for larger mining heights
and less subsidence for deeper longwalls (Galvin, 2016).
To study the possible link of the residual subsidence with the original mining characteristics in
more detail, several groups of locations were selected (Fig. 12). First, three locations were
selected where, underneath, no mining had ever taken place (Table 4a). Second, two locations
with a small amount of mining, i.e., two panels only and with a total mining height of 2 and 2.5
m, respectively (Table 4b). Third, three locations were selected with extensive mining, i.e., 7 or 8
panels and a total mining height of 9.2 to 10.3 m (Table 4c). As for the two locations with
maximum movement (Table 3), the 10 reflectors in the most immediate vicinity were studied. It
was not easy to find an adequate number of locations so proper analyses could be done, i.e.,
enough reflectors had to be present in both observation periods at a close distance, and the same
mining conditions had to exist underneath these reflectors.
When one looks at the average total residual subsidence over the five-year time zone, one gets -
29/-26/-37 mm (no mining), -36/-23 mm (limited mining), and -29/-46/-33 mm (extensive
mining), respectively. Hence, one cannot conclude that the amount of mining underneath a
relatively small area is affecting the residual subsidence, certainly if we also point out that
location $Ext_B$ (-46 mm) was situated very close to the location of the overall maximum of the
residual subsidence in space. When looking at the minima or maxima, also no distinction is
observed between the three groups of the amount of mining. This confirms what was observed
when comparing both locations of maximum movement (Sect. 4.1).



By looking at the data of Tables 4b and 4c as a function of the mining depth, no clear trend is
observed. For the two locations with a limited amount of mining, the most shallow mining
resulted in the largest residual subsidence, while, for the three locations with extensive mining,
the largest residual subsidence was for the deepest exploitation.
When comparing the two locations of maximum movement in Sect. 4.1, there was the possibility
that more residual subsidence occurred directly above the more recent longwall panel. This
would be logical. Therefore, Table 5 classifies the various locations as a function of the most
recent longwall panel underneath. Taking into account the large number of possible parameters
that influenced the results, the trend of these seven locations is indeed that the locations above
the most recent panels resulted in larger residual subsidence. However, it must be pointed out
that the location with the second most recent mining has undergone, on average, less movement
than one of the locations without mining underneath (i.e., No$_C$ with -36.8 mm). So, it certainly
cannot be concluded that there is a simple one-on-one relationship with the time since
exploitation. Two different panels were mined in 1990 and 1992, respectively, but,
unfortunately, no reflectors or insufficient reflectors were present above these panels.
When comparing the residual subsidence in the north-south transect (Fig. 9a) with the map of
longwall panels, one can observe that the zone of influence is larger than expected based on the
normally-used values of the angle of draw. Based on the latter values and the depth of
exploitation, the influence zone during the phase of subsidence should be limited to the zone
between Latitude 50.995°N and 51.06°N, since no exploitation took place any further north than
Latitude 51.05°N in the transect selected or its immediate surroundings or any further south than
Latitude 51.005°N (Van Tongeren and Dreesen, 2004). This was confirmed in the northern part
of this transect. However, as far south as 50.98°N, residual subsidence clearly was observed, i.e.,
1.5 km further away than the theory would predict.
**4.3 Influence of mining characteristics on uplift after closure**
What was explained in previous section is the process that was initiated by the caving process,
and it can be seen as a mechanical stress-deformation process that includes time-dependent
aspects. However, once the underground activities ceased and the underground access was closed
off, including dismantling of the pumping installations, the underground begins to be flooded
(Bekendam and Pöttgens, 1995; Caro Cuenca et al., 2013; de Vent and Roest, 2013). In the
beginning, the water finds its way through various pathways, including open roadways,
permeable faults, and volumes of loose blocks. But there is no reason the rock mass adjacent to
the mined area or between mined areas would not be submerged, and this leads to new processes.
In the literature (Herrero et al., 2012), the swelling of clay minerals of argillaceous rocks under
the influence of water is considered to be the main factor for inducing uplift. Swelling is
governed by the swelling pressure and is, therefore, linked to the mining depth. Caro Cuenca et
al. (2013) showed clearly the direct correlation between the increase of the water level in the
underground areas and the uplift. In all cases, the groundwater levels showed even a very high





correlation (~0.97) with surface displacements. Apart from the uplift, Herrero et al. (2012)
pointed out that, due to the flooding, the mechanical properties of argillaceous rocks are affected
significantly by water, resulting in a decrease of 60 to 80% of their strength, which reactivates
the downward settlement.
For the same average locations, as for the first observation period, the minimum, average, and
maximum uplift of the five-year time zone for 10 reflectors are given in Table 4. By considering
the three groups as a function of the amount of mining, one gets average uplifts of 30/8/46 mm
(no mining), 54/52 mm (limited mining), and 60/43/60 mm (extensive mining), respectively.
Hereby, it must be pointed out that the average value of 8 mm was recorded at the far NE of the
study area, outside the mining area, and at a distance of about 3 km from the location with the
maximum uplift. Although the two smallest of these eight average values were for the group
without mining and the two largest were for the group of extensive mining, one should be very
careful in linking the amount of uplift with the amount of mining directly underneath. Earlier
research also indicated that there is not a clear link between the uplift rate and mining (or the
absence of mining) directly underneath (Vervoort and Declercq, 2016).
Often, one links the largest uplift to zones with the largest subsidence and estimates the total
uplift as 3 to 4% of the total subsidence (Herrero et al., 2012). Bekendam and Pöttgens (1995)
also concluded that, generally, the uplift is 2 to 4% of the subsidence; the latter conclusion is for
the same Campine basin, but above the Dutch coal mines to the east. This cannot be confirmed
by the area studied here and, of course, for the time periods considered; only the residual
subsidence rate is known. As pointed out earlier, no public data were available for the subsidence
that occurred prior to satellite monitoring, but by applying the rule of thumb for estimating the
total subsidence, one could estimate that the subsidence was from about 1.5 to 11 m in the area
studied, and 3% of this would mean that a total of 45 to 330 mm of uplift finally would occur
above the mined out area. If this were correct, then the uplift during the second observation
period (until 2010) would have reached only the bottom part of this predicted range; in other
words, one can still expect more uplift above the mining area and immediate surroundings.
As discussed in Sect. 4.2, the influence zone during the phase of subsidence should be limited to
the zone between Latitudes 50.995°N and 51.06°N. In the northern part of the north-south
transect that was considered, this was confirmed for the residual subsidence, but, in the south, the
influence was about 1.5 km more to the south. For the uplift until 2010 (Fig. 9a), the zone of
influence (e.g., an uplift of more than 5 to 10 mm) corresponded well with the limits of 50.99°N
and 51.06°N. However, after 2010, the extent of the uplift zone could have increased.
Based on all of the information that was collected, there is no indication that the process of uplift
is directly linked to the mining characteristics. It is more likely that the uplift as a result of the
flooding is initiated at or close to the shafts, where most likely the deepest point is situated and
where the pumping station was situated. From that central location, further flooding (in the
horizontal direction) and rise of mine water (in the vertical direction) are extended, creating a
further uplift at that central location and an initiation of uplift further away from the central area.



Of course, the fact that mining and caving have taken place has an effect. It is the main reason
that water flows into the underground workings. However, the local situation (e.g., the depth,
extent, or time of mining) does not seem to have a very significant influence on uplift. When
looking at the interpolated curve of Fig. 9a, no local irregularities are noted; the curve itself also
is very symmetric, much more so than the curve of residual subsidence (Fig. 9b).
**5 Conclusions**
Most research of surface movement above underground mines focuses on the direct effect of
mining, i.e., within the lifetime of the mine, and less attention is given to the long-term impact of
mining on surface movements. As at the end of the last century, several coal basins were closed
in Europe, and researchers began to observe a new phenomenon, i.e., the uplift of the surface as a
consequence of the flooding of the underground workings (Bekendam and Pöttgens, 1995). Also,
cases were reported of damage to buildings and infrastructure during the uplift phase (Baglikow,
2011; de Vent and Roest, 2013; Caro Cuenca et al., 2013). During that period, satellite images
with frequent and detailed measurements of the surface movement over large areas became
available, so this topic could be studied further. To date, the focus has been mainly on
understanding the phenomenon (e.g., Herrero et al., 2012) and identifying general trends, like for
example the link with the rise in the water level (Caro Cuenca et al., 2013; Devleeschouwer et
al., 2008). In this study, the residual subsidence after closure, as well as the initiation and further
evolution of the uplift were investigated for an area of 22 km² above the Houthalen coal mine,
which was closed in 1992. We tried to better quantify the movement after closure and the
difference between the residual downward movement and the ultimate uplift of the surface by
considering past mining directly below the observation points. All this has led to the following
conclusions:
- In the first five years following the closure of the coal mine (between mid-August 1992 and
mid-August 1997), the area studied was still characterized by an overall downward movement;
the average residual subsidence was -33 mm over five years, corresponding to a rate of about -6
mm per year. The maximum rate for the studied area was about -16 mm per year (or a total of -
80 mm).
- Although large residual movements may occur next to small movements, clustering was
present, and it resulted in areas with, on average, smaller residual subsidence and other areas
with larger values; certainly when looking at the north-south sections, there was a clear zone in
which the maximum residual subsidence occurred.
- In absolute terms, the rate of uplift was about the same order of magnitude as the residual
subsidence, but, in fact, it was slightly larger; an average rate of uplift of 9 mm/year was
observed for the period between mid-September 2005 and mid-September 2010, in comparison
to the average rate of -6 mm/year in the five years following the closure.



- The zone in which the maximum uplift occurred was clearly at a different location from the
zone with the maximum residual subsidence.
- The curve of the uplift along the north-south sections was very symmetric, which was not the
case for the curve of the residual subsidence.
- There was no clear sign that the amount of mining underneath a relatively small area had an
effect on the residual subsidence. However, there was some indication that the locations above
the most recent panels resulted in larger residual subsidence values. There is not a simple one-
on-one relationship with the time since exploitation. The zone of influence was larger than one
would expect based on the normally-used values of the angle of draw and depth of mining.
- Based on all of the information that was collected, there was no indication that the process of
uplift was directly linked to the mining characteristics. It is more likely that the uplift as a result
of flooding was initiated at or close to the shafts; from that central location, the additional
flooding (in the horizontal direction) and rise of mine water (in the vertical direction) were
extended, creating additional uplift at that central location and initiating uplift further away from
the central area.
Most concepts that one finds in textbooks dealing with surface subsidence above longwalls
considers either the impact of mining a single panel or a relatively-simple mining geometry
and/or mining sequence (e.g., mining a single seam with adjacent panels, which are mined in a
successive sequence). The latter is certainly typical for several countries, including the large coal
producers, such as Australia, South Africa, and the USA. In Europe, the situation was often
different. For the mine studied, a total of 10 seams were partly mined over a time period of 60
years (between 1932 and 1992) at depths varying from 539 to 967 m. However, the situation was
not significantly different when a shorter time is considered. For example, in the 1970s, seven
seams still were being mined at depths varying from 556 to 824 m. As one also observes on the
map of longwall panels (Fig. 1), there was no systematic geometry or a systematic approach of
mining the different panels. These observations probably explain why no clear link has been
established between mining characteristics and residual subsidence. The entire area was rather in
movement. For the amount of uplift, such one-on-one relationships were nonexistent. As
illustrated above, one can best visualize the uplift as starting at or close to the shafts, whereby a
further uplift occurred in the following years at that central location, and uplift was initiated
farther away from this central area. This seems to be in accordance with the process of flooding
the underground and the systematic rise of the water level. It will be interesting to investigate the
further evolution of the uplift, when more recent satellite data become available.
The process of subsidence and the one of uplift are entirely different. The first is caused by a
caving process and is mainly a mechanical stress-deformation process, including time-dependent
aspects, while the process of uplift is caused by the swelling of the clay-minerals in the rocks,
due to flooding. Hence, one cannot assume that the areas where one has the greatest risk for
damage to infrastructure due to subsidence are the same areas for the hazards linked to the uplift
process.




*Acknowledgements.* The input by Pierre-Yves Declercq from the Geological Survey of Belgium,
Royal Belgian Institute of Natural Sciences, Brussels is greatly acknowledged for providing the
necessary basic data on surface movements and mining characteristics.

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





**List of Figures**


**Fig. 1.** Map of longwall panels in area studied, i.e. between a Latitude of 51.01°N and 51.05°N,
and between a Longitude of 5.33°E and 5.40°E.

**Fig. 2.** Distribution of total surface movement (in mm): a. Period 1, 5-year time zone, from mid-
August 1992 through mid-August 1997; b. Period 1, 2.5-year time zone, from July 1998 through
December 2000. Largest subsidence (negative values) is plotted to the right.

**Fig. 3.** Spatial variation of total surface movement in the area between Latitude 51.01°N and
51.05°N, and Longitude 5.33°E and 5.40°E: a. Period 1, 5-year time zone, from mid-August
1992 through mid-August 1997 (all reflectors; color scale is in mm); b. Period 1, 2.5-year time
zone, from July 1998 through December 2000 (only reflectors with a upward movement during
this time zone).

**Fig. 4.** Variation of the total surface movement along a north-south transect, situated for
Longitude between 5.37°E and 5.38°E: a. Period 1, 5-year time zone, from August 1992 through
August 1997; b. Period 1, 2.5-year time zone, from July 1998 through December 2000.

**Fig. 5.** Distribution of total surface movement (in mm): a. Period 2, 5-year time zone, from mid-
September 2005 through mid-September 2010; b. Period 2, 2.5-years time zone, from December
2003 through June 2006. Largest uplift (positive values) is plotted to the right.

**Fig. 6.** Spatial variation of total surface movement in the area between Latitude 51.01°N and
51.05°N, and Longitude 5.33°E and 5.40°E: a. Period 2, 5-year time zone, from mid-September
2005 through mid-September 2010 (all reflectors; color scale is in mm); b. Period 2, 2.5-year
time zone, from December 2003 through June 2006 (only reflectors with a downward movement
during this time zone).

**Fig. 7.** Variation of the total surface movement along a north-south transect, situated for
Longitude between 5.37°E and 5.38°E: a. Period 2, 5-year time zone, from mid-September 2005
through mid-September 2010; b. Period 2, 2.5-year time zone, from December 2003 through
June 2006.

**Fig. 8.** Variation of the total surface movement along a east-west transect, situated for Latitude
between 51.018°N and 51.026°N for the 5-year time zone in Period 2 (from mid-September 2005
through mid-September 2010).



**Fig. 9.** Smoothed curves fitted for the total surface movement along a north-south transect,
situated for Longitude between 5.37°E and 5.38°E: a. The two 5-year time zones (see
respectively Figs. 4a and 7a); b. The two 2.5-year time zone (see respectively Figs. 4b and 7b).

**Fig. 10.** Evolution of subsidence over 5-year time zone in first observation period (from mid-
August 1992 through mid-August 1997): a. 10 reflectors around coordinates 51.036°N, 5.375°E
(Location Max $_{\text{RES SUBS}}$); 10 reflectors around coordinates 51.022°N, 5.377°E (Location Max
$_{\text{UPLIFT}}$).

**Fig. 11.** Evolution of uplift over 5-year time zone in first observation period (from mid-
September 2005 through mid-September 2010): a. 10 reflectors around coordinates 51.036°N,
5.375°E (Location Max $_{\text{RES SUBS}}$); 10 reflectors around coordinates 51.022°N, 5.377°E (Location
Max $_{\text{UPLIFT}}$).

**Fig. 12.** Indication of selected locations on map of exploitation panels in area studied (between
Latitude of 51.01°N and 51.05°N, and between Longitude 5.33°E and 5.40°E) to study link with
mining characteristics.





**List of Tables**



**Figures**

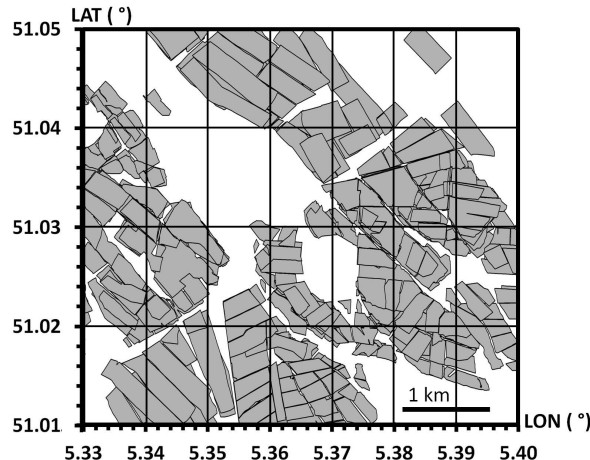

**Fig. 1.** Map of longwall panels in area studied, i.e. between a Latitude of 51.01°N and 51.05°N, and between a Longitude of 5.33°E and 5.40°E.

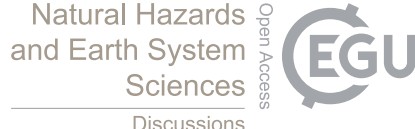



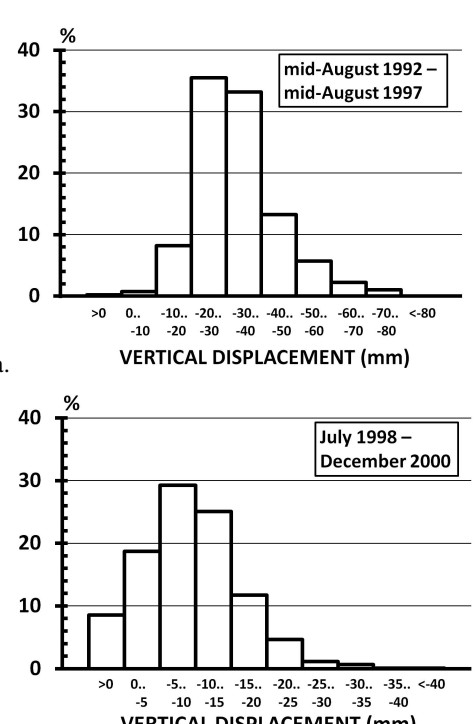

**Fig. 2.** Distribution of total surface movement (in mm): a. Period 1, 5-year time zone, from mid-August 1992 through mid-August 1997; b. Period 1, 2.5-year time zone, from July 1998 through December 2000. Largest subsidence (negative values) is plotted to the right.



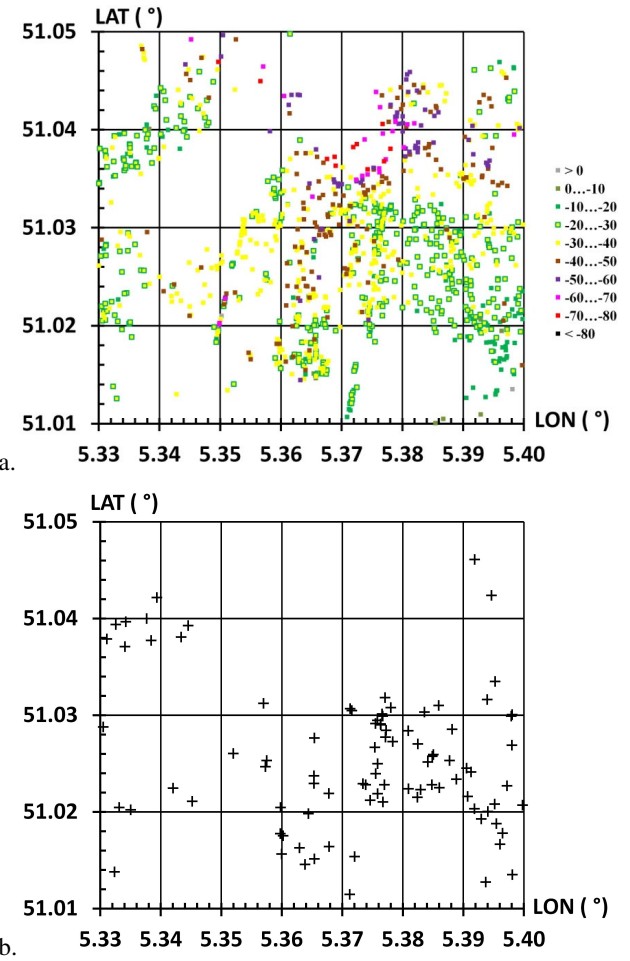

**Fig. 3.** Spatial variation of total surface movement in the area between Latitude 51.01°N and 51.05°N, and Longitude 5.33°E and 5.40°E: a. Period 1, 5-year time zone, from mid-August 1992 through mid-August 1997 (all reflectors; color scale is in mm); b. Period 1, 2.5-year time zone, from July 1998 through December 2000 (only reflectors with a upward movement during this time zone).





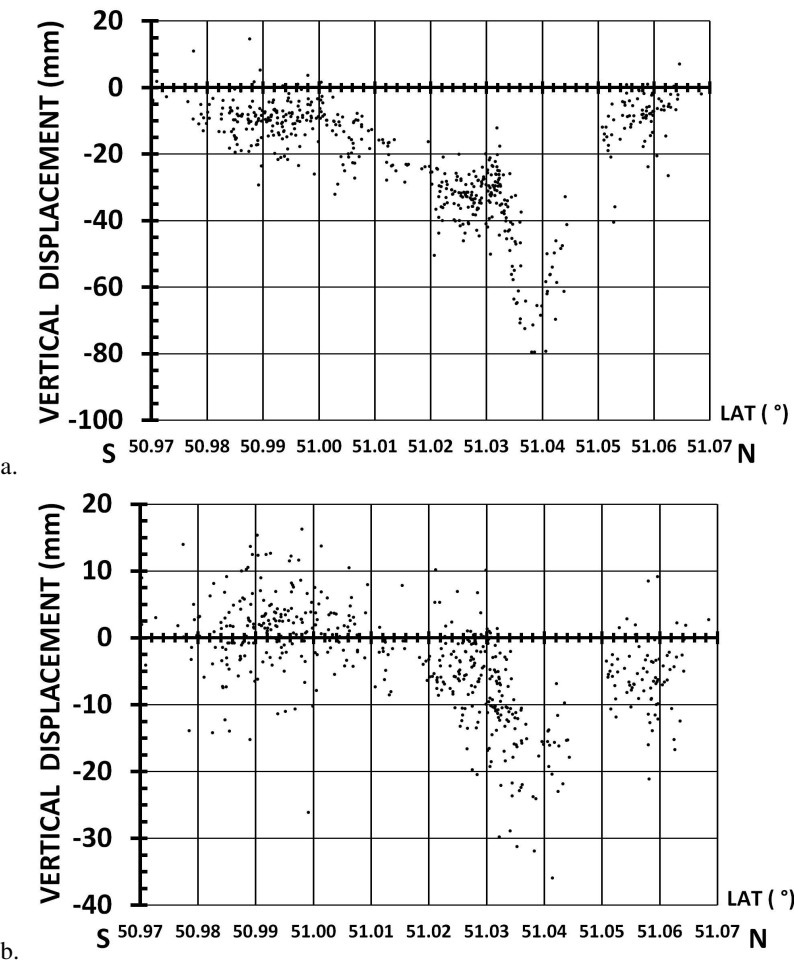

**Fig. 4.** Variation of the total surface movement along a north-south transect, situated for Longitude between 5.37°E and 5.38°E: a. Period 1, 5-year time zone, from August 1992 through August 1997; b. Period 1, 2.5-year time zone, from July 1998 through December 2000.



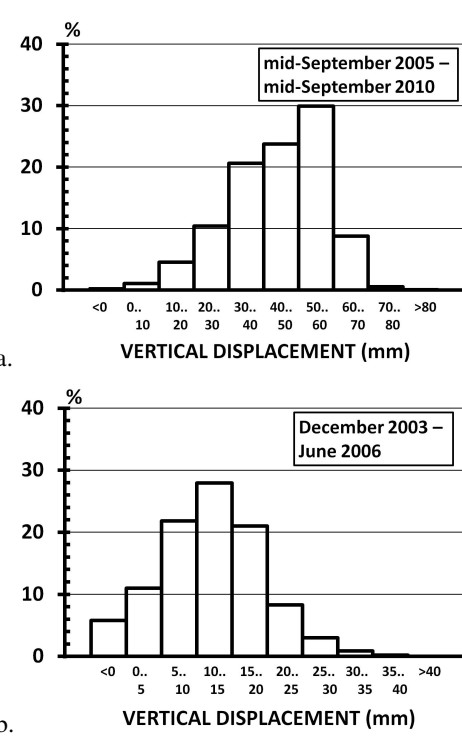

**Fig. 5.** Distribution of total surface movement (in mm): a. Period 2, 5-year time zone, from mid-September 2005 through mid-September 2010; b. Period 2, 2.5-years time zone, from December 2003 through June 2006. Largest uplift (positive values) is plotted to the right.



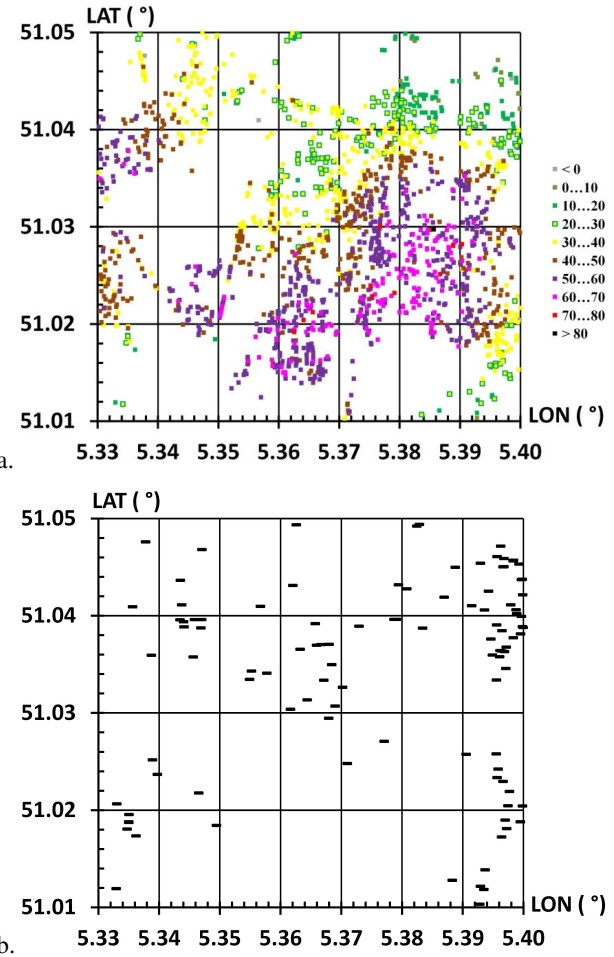

**Fig. 6.** Spatial variation of total surface movement in the area between Latitude 51.01°N and 51.05°N, and Longitude 5.33°E and 5.40°E: a. Period 2, 5-year time zone, from mid-September 2005 through mid-September 2010 (all reflectors; color scale is in mm); b. Period 2, 2.5-year time zone, from December 2003 through June 2006 (only reflectors with a downward movement during this time zone).


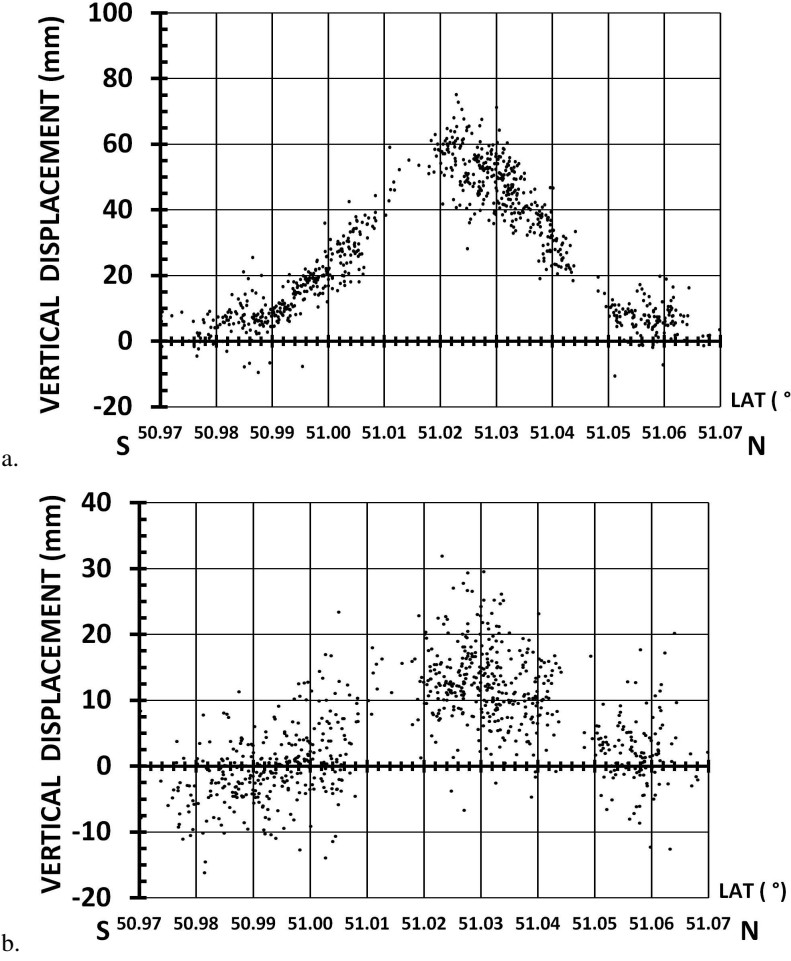

**Fig. 7.** Variation of the total surface movement along a north-south transect, situated for Longitude between 5.37°E and 5.38°E: a. Period 2, 5-year time zone, from mid-September 2005 through mid-September 2010; b. Period 2, 2.5-year time zone, from December 2003 through June 2006.





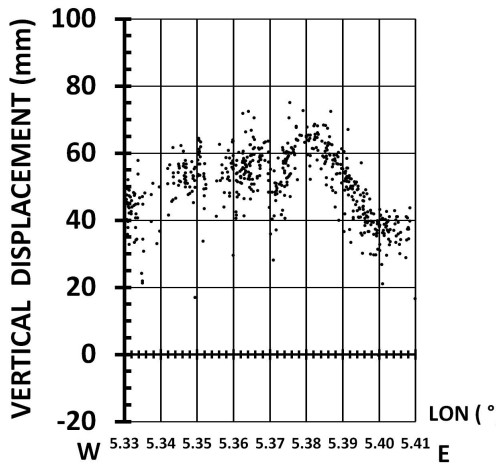

**Fig. 8.** Variation of the total surface movement along a east-west transect, situated for Latitude between 51.018°N and 51.026°N for the 5-year time zone in Period 2 (from mid-September 2005 through mid-September 2010).



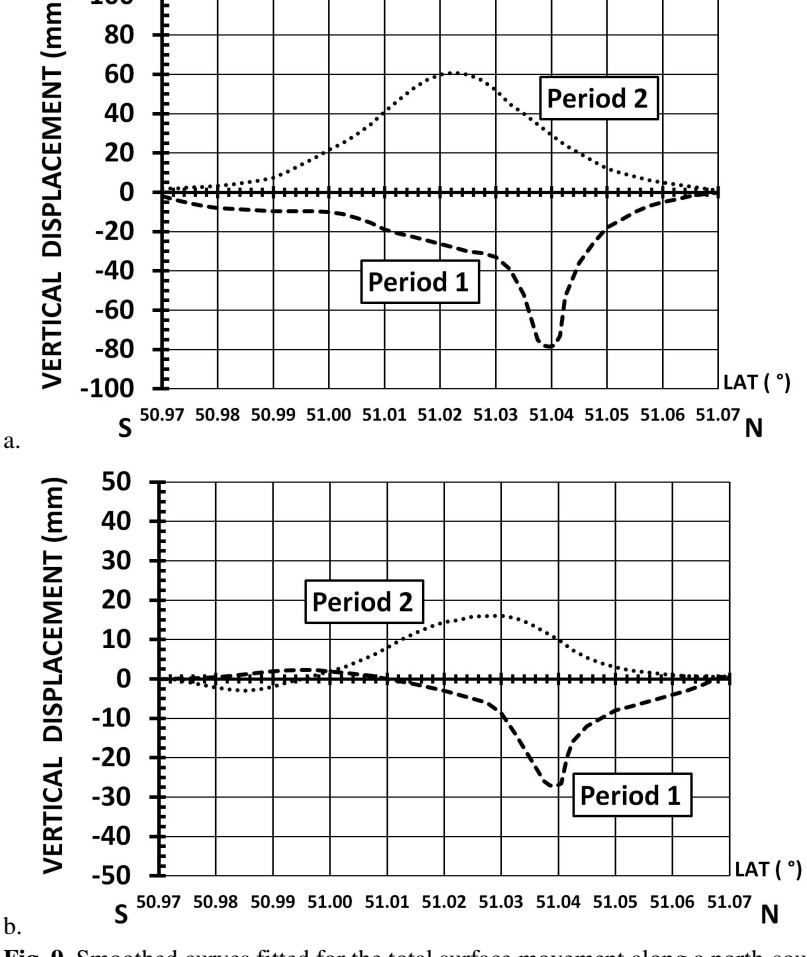

**Fig. 9.** Smoothed curves fitted for the total surface movement along a north-south transect, situated for Longitude between 5.37°E and 5.38°E: a. The two 5-year time zones (see respectively Figs. 4a and 7a); b. The two 2.5-year time zone (see respectively Figs. 7b and 7b).



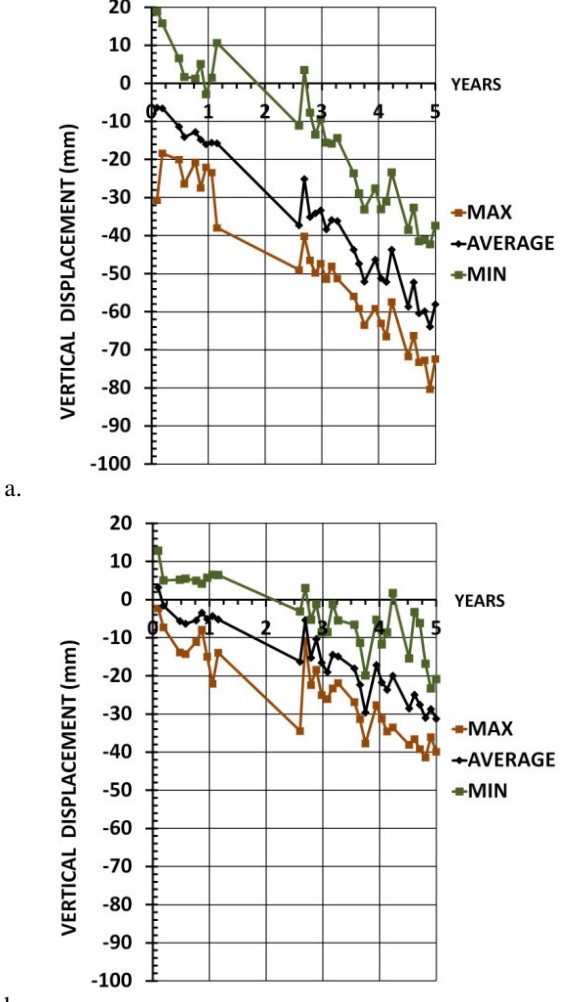

**Fig. 10.** Evolution of subsidence over 5-year time zone in first observation period (from mid-August 1992 through mid-August 1997): a. 10 reflectors around coordinates 51.036°N, 5.375°E (Location Max $_{RES\ SUBS}$); 10 reflectors around coordinates 51.022°N, 5.377°E (Location Max $_{UPLIFT}$).



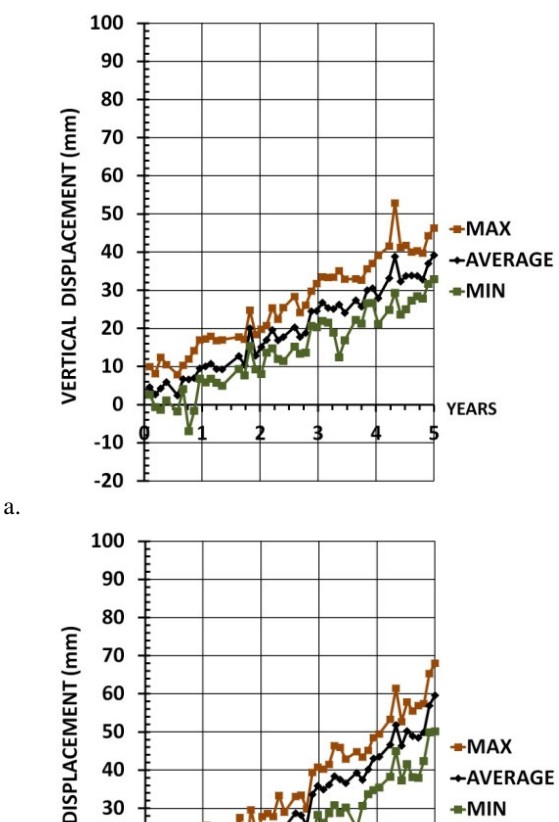

a.

b.

**Fig. 11.** Evolution of uplift over 5-year time zone in first observation period (from mid-September 2005 through mid-September 2010): a. 10 reflectors around coordinates 51.036°N, 5.375°E (Location Max $_{RES\ SUBS}$); 10 reflectors around coordinates 51.022°N, 5.377°E (Location Max $_{UPLIFT}$).





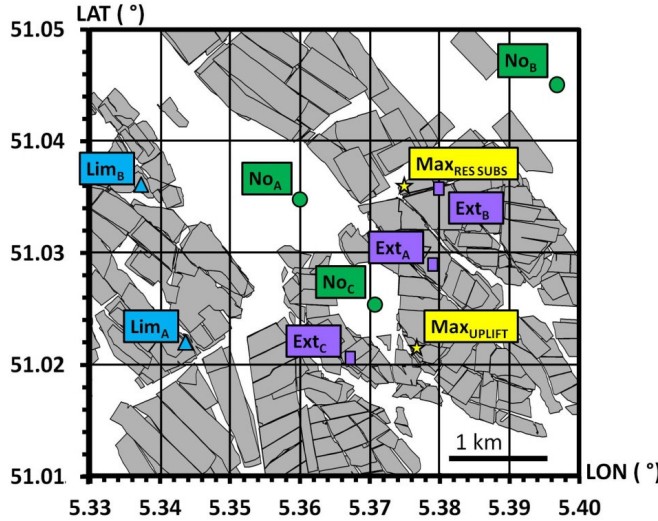

**Fig. 12.** Indication of selected locations on map of exploitation panels in area studied (between Latitude of 51.01°N and 51.05°N, and between Longitude 5.33°E and 5.40°E) to study link with mining characteristics.





**Tables**

**Table 1.** Information on total surface movement during the two times zones of 5 and 2.5 years considered in the first observation period for the total area studied.

|  | Period 1, 5-year time zone, mid-August 1992- mid-August 1997 | Period 1, 2.5-year time zone, July 1998- December 2000 |
|---|---|---|
| Number of reflectors | 1,073 | 1,073 |
| Minimum | 5.8 mm | 14.9 mm |
| Average | -32.9 mm | -9.0 mm |
| Maximum | -79.5 mm | -40.5 mm |
| Standard Deviation | 11.8 mm | 7.2 mm |
| Skewness (*) | 0.83 | 0.24 |

(*) Positive skewness means a long tail for large values, i.e. large subsidence in the first observation period.



**Table 2.** Information on total surface movement during the two times zones of 5 and 2.5 years considered in the second observation period for the total area studied.

|  | Period 2, 5-year time zone mid-September 2005 - mid-September 2010 | Period 2, 2.5-year time zone December 2003 - June 2006 |
| --- | --- | --- |
| Number of reflectors | 1,808 | 1,808 |
| Minimum | -9.9 mm | -21.7 mm |
| Average | 43.9 mm | 11.8 mm |
| Maximum | 83.5 mm | 37.1 mm |
| Standard Deviation | 13.8 mm | 7.6 mm |
| Skewness (*) | -0.58 | -0.19 |

(*) Negative skewness means a long tail for small values, i.e. small uplift in the second observation period.





**Table 3.** Information on the two locations, corresponding to the zones with approximately largest residual subsidence in first period (Max $_{RES\ SUBS}$) and largest uplift in second observation period (Max $_{UPLIFT}$): information on total movement of 10 reflectors around coordinates given in 5-year time zones studied and data on exploitation below these locations.

| LOCATION | | Max $_{RES\ SUBS}$ | Max $_{UPLIFT}$ |
|---|---|---|---|
| Coordinates, | LAT | 51.036°N | 51.022°N |
| | LON | 5.375°E | 5.377°E |
| Vertical movement over 5 years: | | | |
| first period: | MIN | -37.4 mm | -20.9 mm |
| | AVERAGE | -58.1 mm | -31.2 mm |
| | MAX | -72.4 mm | -40.0 mm |
| second period: | MIN | 33.0 mm | 50.2 mm |
| | AVERAGE | 39.3 mm | 59.7 mm |
| | MAX | 46.3 mm | 68.0 mm |
| Exploitation: | | | |
| NUMBER OF LONGWALLS | | 4 | 6 |
| OLDEST YEAR | | 1968 | 1939 |
| MOST RECENT YEAR | | 1982 | 1959 |
| MIN DEPTH | | 686 m | 565 m |
| MAX DEPTH | | 796 m | 712 m |
| TOTAL MINING HEIGHT | | 7.0 m | 9.3 m |





**Table 4.** Information of selected locations, i.e. movement of 10 reflectors around coordinates given over 5-year time zones in both observation periods and mining characteristics underneath locations: a. No exploitation; b. Limited exploitation; c. Extensive exploitation.

a.

| LOCATION | | $No_A$ | $No_B$ | $No_C$ |
|---|---|---|---|---|
| Coordinates, | LAT | 51.035° | 51.045° | 51.025° |
| | LON | 5.360° | 5.397° | 5.371° |
| Vertical movement over 5 years: | | | | |
| first period: | MIN | -24.3 mm | -14.5 mm | -27.9 mm |
| | AVERAGE | -28.7 mm | -25.7 mm | -36.8 mm |
| | MAX | -34.6 mm | -40.4 mm | -46.0 mm |
| second period: | MIN | 22.9 mm | -6.6 mm | 28.2 mm |
| | AVERAGE | 29.8 mm | 8.4 mm | 45.6 mm |
| | MAX | 42.5 mm | 24.6 mm | 54.8 mm |
| Exploitation: | | None | None | None |

b.

| LOCATION | | $Lim_A$ | $Lim_B$ |
|---|---|---|---|
| Coordinates, | LAT | 51.022° | 51.036° |
| | LON | 5.344° | 5.337° |
| Vertical movement over 5 years: | | | |
| first period: | MIN | -27.9 mm | -10.1 mm |
| | AVERAGE | -35.7 mm | -22.7 mm |
| | MAX | -42.3 mm | -33.7 mm |
| second period: | MIN | 46.7 mm | 41.7 mm |
| | AVERAGE | 54.0 mm | 52.2 mm |
| | MAX | 62.3 mm | 61.9 mm |
| Exploitation: | | | |
| NUMBER OF LONGWALLS | | 2 | 2 |
| OLDEST YEAR | | 1954 | 1933 |
| MOST RECENT YEAR | | 1977 | 1938 |
| MIN DEPTH | | 613 m | 688 m |
| MAX DEPTH | | 736 m | 743 m |
| TOTAL MINING HEIGHT | | 2.5 m | 2.0 m |





c.

| LOCATION | | Ext$_A$ | Ext$_B$ | Ext$_C$ |
|---|---|---|---|---|
| Coordinates, | LAT | 51.029° | 51.036° | 51.021° |
| | LON | 5.379° | 5.380° | 5.367° |
| Vertical movement over 5 years: | | | | |
| first period: | MIN | -22.8 mm | -24.9 mm | -26.8 mm |
| | AVERAGE | -28.9 mm | -45.9 mm | -32.8 mm |
| | MAX | -34.7 mm | -79.5 mm | -50.4 mm |
| | | | | |
| second period: | MIN | 51.1 mm | 35.8 mm | 48.9 mm |
| | AVERAGE | 59.6 mm | 43.3 mm | 59.7 mm |
| | MAX | 67.6 mm | 48.9 mm | 70.5 mm |
| Exploitation: | | | | |
| NUMBER OF LONGWALLS | | 7 | 8 | 7 |
| OLDEST YEAR | | 1941 | 1943 | 1947 |
| MOST RECENT YEAR | | 1968 | 1971 | 1965 |
| MIN DEPTH | | 633 m | 629 m | 585 m |
| MAX DEPTH | | 888 m | 965 m | 735 m |
| TOTAL MINING HEIGHT | | 9.9 m | 10.3 m | 9.2 m |





**Table 5.** Information on residual subsidence of the locations considered in Table 3 and Table 4, re-ordered as a function of the most recent exploitation panel.

| Most recent year of exploitation | Minimum residual subsidence | Average residual subsidence | Maximum residual subsidence | Location |
|---|---|---|---|---|
| 1938 | -10.1 mm | -22.7 mm | -33.7 mm | $Lim_B$ |
| 1959 | -20.9 mm | -31.2 mm | -40.0 mm | $Max_{UPLIFT}$ |
| 1965 | -26.8 mm | -32.8 mm | -50.4 mm | $Ext_C$ |
| 1968 | -22.8 mm | -28.9 mm | -34.7 mm | $Ext_A$ |
| 1971 | -24.9 mm | -45.9 mm | -79.5 mm | $Ext_B$ |
| 1977 | -27.9 mm | -35.7 mm | -42.3 mm | $Lim_A$ |
| 1982 | -37.4 mm | -58.1 mm | -72.4 mm | $Max_{RES\ SUBS}$ |