# Peer review of "Surface movement above an underground coal longwall mine after closure"

_Natural Hazards and Earth System Sciences, 2016_

## Referee Comment (RC1) · Anonymous Referee #1 · 2 Jun 2016

The presented topic is of high significance. It addresses environmental impacts of underground coal mining after closure in terms of residual subsidence and uplift and contributes as such to after mine care and safety. The paper is well presented and structured, includes a thorough analysis of spatial data leading to new insights into mechanisms controlling vertical ground movements after mine closure.

From the reviewer's point of view, view minor changes may support the contribution:

1) It is advised to use passive voice consistently in the text, no use of we 2) Lines 143 – 146 appear to the reader a bit unclear (overlap in time zones) – this could be rewritten more clearly 3) At the beginning of section 3 the reader may already ask himself about the sequence of mining as it may likely influence the timely subsidence/uplift behaviour. Although discussed later in section 4, it may help to introduce basic mining parameters

(depth, thickness, and mining sequence) already in the introduction section next to the description of geology. A map containing years of production of the panels may help visualising a possible link between ground movement and extraction sequence. 4) Sentence in line 185 related to EW section could be left out (Ideally, . . .), since no section is presented. 5) Formulation in line 263 appears unfortunate: Better: 'It becomes obvious that an uplift over the entire area took place' 6) General question on section 3: Does the author suspect that when using different time periods (e.g. 2010 – 2013 or before 1992) results may differ, especially with respect to the rate of residual subsidence and uplift, e.g. due to the extraction sequence? In other words, is the comparison between 2 x 5 year periods just a snapshot or can observations be generalized? A short discussion may address this issue (partly later answered in section 4.3). 7) Interpretation lines 356-359: Note that especially in the centre of the field, considered the x and y coordinates for min and extensive mining activities are rather close. From the map the distance appears to be less than the area of influence. Movements may be superimposed and the ground movement behaviour at a min location is most likely affected by the subsidence from a neighbouring panel. Thus the ground movements of points which are close to each other may not interpreted independently. Conclusions drawn can be associated with some fuzziness, which would reflect the limit of this study by not taking the spatial nature (i.e. angel of draw of 45 degrees and superimposition of effect of multiple panels) into account. 8) Conclusions: One main conclusion is that there is obviously no one-to-one relation between some parameters and the effects of residual and uplift. A suggestion may be added to further research on this multi-variate problem using techniques including unsupervised learning and supervised learning. 9) Note on line 492: Much research has been conducted and published (mostly in German or Polish language) on complex mining geometries and multi-seam mining in the Polish and German hard coal industry by e.g. H. Kratzsch or A. Sroka. Only some references are available in English language.

[Figure]

2016.

---

## Referee Comment (RC2) · Anonymous Referee #2 · 15 Jun 2016

General comments Very interesting article, especially since the issue of the residual subsidence and the uplift of the area after mining activity in Europe is increasingly important due to the fairly widespread liquidation of active underground European mining. The paper address relevant scientific and technical questions within the scope of NHESS. The paper present new data and results. There is up to international standards. The methods and assumptions are valid and clearly outlined. The results are sufficient to support the interpretations and the conclusions. The author reaches substantial conclusions. The description of the data, the method and the results obtained is sufficiently complete and accurate to allow their reproduction by fellow scientists. The title clearly and unambiguously reflect the contents of the paper. The abstract provide a concise, complete and unambiguous summary of the work and the obtained results. The title and the abstract are pertinent, and easy to understand to a wide and

diversified audience. The overall presentation is well structured, clear and easy to understand by a wide and general audience. The length of the paper is adequate. The technical language is precise and understandable by fellow scientists. I am not English but in my opinion the English language is of good quality, fluent, simple and easy to read and understand by a wide and diversified audience. Specific comments 1) It is interesting how accurate is the method of interferometry especially for just such analysis. It would be interesting, if possible, to compare the measurement results with the results of measurements of the classical levelling method. 2) Conclusions are interesting but also intuitive. If uplift is associated with swelling of clay minerals that should be linked to the phenomenon of rising water levels and the occurrence of these minerals in the geological layers. As noted in an article in Carboniferous there are no clay layers. These occur in the overburden. Therefore, uplift should be associated with the liquidation of depression cone throughout its previous range, bevor mining activity has been finished. Uplift should occur actually in the area where there is no mined out coal seams - that is just around the shaft because there is still a safety pillar of the vertical shaft. The caving above exploited coal seams serve as ways of spread of water. Therefore, one should not expect the uplift especially above areas of former mining operation. Subsidence and uplift there are independent phenomena, only slightly linked by the mining operation. This is somewhat due to the paper, and was confirmed by the Author of the article. 3) It would be interesting to analyse the uplift of the ground in respect to the rising of water levels in different aquifers. 4) The phenomenon of the residual subsidence is time dependent, it is obvious and has been stressed in the article. Therefore, it is difficult to assess real residual subsidence above the area of different mine panels, each of which ended its activity at different periods of time. On the contrary, the assessment of uplift is associated with a rise of the water and it can be well assessed after 1992 when pumping of the mine water has been finished. 5) It is not easy to understand the sequence of Figures 3a and 3b, and 6a and 6b. In Fig. 3a and 3b the phenomena are presented chronologically and 6a/6b contrariwise, it is very difficult to interpret. Perhaps it makes sense but I have not found a justification

for such order. 6) In the list of references there is no one position of Polish literature. Knothe theory, which was crucial for the prediction of subsidence has been recalled from Chinese literature, what is curious (587-589).

Technical correction 1) Figures 3 and 6 are difficult to read. Probably the coloured symbols would be better. And perhaps Author should use some other symbols to display the different settlement and uplift classes? 2) To assess whether there is a relationship between the operation and the residual subsidence and uplift in Figures 3 and 6 the contours of operation should be shown. Without this the assessment is very difficult; 3) On the Figures 1, 3 and 6 there is no section line drawn. It is a pity, because it would make easier the interpretation of the results presented on Figures 4, 7 and 8;

———————————————————

---

## Author Comment (AC1) · 5 Jul 2016

nhess-2016-134 Author comments on Referees comments for paper "Surface movement above an underground coal longwall mine after closure", A. Vervoort

Dear Editor First of all, I would like to thank both referees for their comments and suggestions (no other comments were posted by the scientific community). Overall, I consider their reviews as positive. Below I write my reaction, point per point, on their points of criticism and on their suggestions. Mostly, I agree with them and I have made already the necessary changes in the manuscript (see file attached), although this was not required yet. I have used the following color code: the text by referees is in black; my comments are in blue and the changes made in the manuscript are in red. (see attached pdf for color version)

[Figure]

I hope that you agree with the changes that I have made. However, if you would still like further improvements, please let me know.

Kind regards, André Vervoort

Anonymous Referee #1

The presented topic is of high significance. It addresses environmental impacts of underground coal mining after closure in terms of residual subsidence and uplift and contributes as such to after mine care and safety. The paper is well presented and structured, includes a thorough analysis of spatial data leading to new insights into mechanisms controlling vertical ground movements after mine closure. I thank the first referee for the positive global evaluation.

From the reviewer's point of view, view minor changes may support the contribution: 1) It is advised to use passive voice consistently in the text, no use of we I know that this point is often a point of discussion and that some journals prefer the passive form and others the active one. I have checked some papers in nhess and I see both forms being published. So, I have made in the revised version no changes to the manuscript on this point. However, if the editor would insist on the passive form, please let me know and I will the necessary changes.

2) Lines 143 – 146 appear to the reader a bit unclear (overlap in time zones) – this could be rewritten more clearly I accept this suggestion and I have reformulated this part, so that it is better formulated. The new text is as follows: In a similar way, it was observed that, at the start of the second period of observation (from December 2003 through October 2010), certain reflectors were still undergoing downward movement. Therefore, in first instance, we looked at 5- year time zones in each observation period, which can be considered to be characterized by a pure downward movement (for the first observation period from mid-August 1992 through mid-August 1997) or a pure up-wards movement (for the second observation period from mid-September 2005 through mid-September 2010). The remaining part of each observation period was also studied

and for comparison purposes, a length of 2.5 years was chosen, i.e. the last 2.5 years of the first period and the first 2.5 years of the second period. These two 2.5-year time zones were from July 1998 through December 2000 and from December 2003 through June 2006, respectively. As the total first observation period was longer than 7.5 years and the second shorter, there was a gap between the time zones of 5 and 2.5 years in the first period and a small overlap in the second period, but the main advantage of doing so was that all time zones could be compared more easily.

3) At the beginning of section 3 the reader may already ask himself about the sequence of mining as it may likely influence the timely subsidence/uplift behaviour. Although discussed later in section 4, it may help to introduce basic mining parameters (depth, thickness, and mining sequence) already in the introduction section next to the description of geology. A map containing years of production of the panels may help visualising a possible link between ground movement and extraction sequence. I have tried to build the paper in a systematic order, whereby the analysis of the data on the surface movement is the central and most important part, followed by an interpretation of the observations. On the other hand, the basic mining parameters were already mentioned in the introduction: At a certain X-Y position within the mined area, one to eight different coals seams were mined. The combined mining height of the several seams varied from 2.0 to 12.3 m within this area. The height of the mining of individual panels varied from 0.9 to 2.7 m, and, normally, about 10 to 40 cm of it were layers of waste rock. In some cases, either no waste rock was mined or only a few centimeters were mined, but, in other cases, almost 1 m of waste rock was recorded as having been mined. As the map indicates, certain zones were not mined. Apart from the zone around the vertical shafts (around the coordinates of Latitude 51.025°N and Longitude 5.370°E), these unmined zones mainly were areas around faults. The latter were composed of a predominant set of NNW - SSE striking normal faults with subordinate N-S to NE-SW striking thrust faults. In the later decades of production, a typical longwall panel had dimensions of 200 by 800 to 1,000 m. The main and tail gates were immediately adjacent to the panel, and they were just single tunnels with a horseshoe

cross-section. So, no barrier or remnant pillars existed between the longwall panels. In the area that we studied, the mining depth varied from 539 to 967 m, and the mining occurred between 1932 and 1992. However, most of the panels were mined in the 1960s and 1970s. In Sect. 4, more details of the mining characteristics are provided, and their possible influences on the surface movements are discussed. So, I would like to keep it this way. Regarding the remark on adding the year of production to each individual panel on Fig. 1: I understand the referee, but I have tried this before but this leads to, on the one hand, an overloaded map with too much information and, on the other hand, not all information can be made visible. The main reason is that up to 8 panels are mined above each other. In fact one needs to see it in 3D. So, the presentation of the information in Tables (see Table 3 and 4) seems me to be the best option, although it is less visual.

4) Sentence in line 185 related to EW section could be left out (Ideally, : : :), since no section is presented. The word 'section' has been replaced and the sentence starting with 'Ideally' has been taken out. The new text is as follows: To better visualize the map of reflectors, the movement along a north-south line is presented in Fig. 4a. To have a sufficient amount of reflectors along this line, a north-south zone was selected for Longitude between 5.37°E and 5.38°E, about 700 m wide in the east-west direction.

5) Formulation in line 263 appears unfortunate: Better: 'It becomes obvious that an uplift over the entire area took place' I assume the referee refers to line 213. I have changed the sentence as suggested. In the five-year time zone from mid-September 2005 through mid-September 2010 at the end of the second observation period, it becomes obvious that an uplift over the entire area took place (Fig. 5a).

6) General question on section 3: Does the author suspect that when using different time periods (e.g. 2010 – 2013 or before 1992) results may differ, especially with respect to the rate of residual subsidence and uplift, e.g. due to the extraction sequence? In other words, is the comparison between 2 x 5 year periods just a snapshot or can observations be generalized? A short discussion may address this issue (partly

later answered in section 4.3). This is an interesting and important point that the referee raises. The period of 5 years is not chosen at random. It is based on earlier research (see Vervoort and Declercq, 2016), in which annual increases in movement were studied in detail. This showed that during both periods of about 5 years there was either only subsidence (first observation period), or only uplift (second observation period).So, that means that if we would have chosen 4 years instead of 5 years, we would have also a pure downward or upward movement, but this would be not the case by choosing 6 or 7 year time zones. In the latter cases, the movement would change direction during the entire period (at least at a significant amount of reflectors). The first sentence of section 3 has been changed and the changes made following comment 2 (see above) should also help to clarify this point. Earlier research (Vervoort and Declercq, 2016) looked at annual increases in surface movement. For your information, the paper Vervoort and Declercq, 2016 is in the mean time accepted for publication. This has also been indicated in the list. Vervoort, A., and Declercq, P.-Y.: Surface movement above old coal longwalls after mine closure. Accepted (May 6th 2016) for publication in Int. J. of Min. Sc. and Techn., 2016.

7) Interpretation lines 356-359: Note that especially in the centre of the field, considered the x and y coordinates for min and extensive mining activities are rather close. From the map the distance appears to be less than the area of influence. Movements may be superimposed and the ground movement behaviour at a min location is most likely affected by the subsidence from a neighbouring panel. Thus the ground movements of points which are close to each other may not interpreted independently. Conclusions drawn can be associated with some fuzziness, which would reflect the limit of this study by not taking the spatial nature (i.e. angel of draw of 45 degrees and superimposition of effect of multiple panels) into account. I fully agree with the comments by the referee. If one would consider the zone of influence of all panels, nearly all points within the area studied are within the zone of influence of at least one panel, of course sometimes a very old panel. An angle of draw of 45° means that the extent of the zone of influence beyond the edge of the panel is the depth of the panel. By

considering the maximum depth at a certain location, the zones of influence are about 700 to nearly 1,000 m wider than the panels (e.g. on Fig. 1). On the other hand, theory tells us that the maximum subsidence is of course larger above the mined out area than at the edges of the zone of influence. In one way, this is also logic. However, by looking at residual subsidence this cannot be clearly concluded, which is one the main findings of this research. To address the point raised by the referee, a sentence has been added in Sect. 4.2: By looking at the map of the exploitation (Fig. 12), this means that nearly the entire area studied is within the zone of influence of at least one longwall panel. Also, earlier (section 4.1), I have added the following sentences: Table 3 provides a summary of some basic information on the exploitation just underneath the two locations. The surface movement is of course not only affected by the mining directly below, but also by the mining around the locations. For an angle of draw of 45° the extent of the zone of influence is even equal to the depth of mining. However, the impact decreases when moving away from a panel, which justifies considering the exploitation in the immediate vicinity. This point is further addressed under comment 8, below. It would also be worthwhile to conduct in future research a similar investigation as the one presented, but whereby data are public on the initial subsidence too, so that a comparison can be made between initial subsidence, residual subsidence, uplift and mining parameters.

8) Conclusions: One main conclusion is that there is obviously no one-to-one relation between some parameters and the effects of residual and uplift. A suggestion may be added to further research on this multi-variate problem using techniques including un-supervised learning and supervised learning. This is a very good suggestion and the following sentence and reference has been added in the Conclusions: As for this area, no one-on-one relationships could be clearly indentified between the surface move-ment and the mining characteristics, future research of this multi-variate problem could benefit by using techniques including unsupervised learning and supervised learning (Noack et al., 2014). Best would be to have data on the initial subsidence, the residual subsidence and the uplift, combined with data on the mining characteristics.

Noack, S., Knobloch, A., Etzold, S.H., Barth, A., and Kallmeier, E.: Spatial predictive mapping using artificial neural networks. In: Proceedings of the International Archives of the Photogrammetry, Remote Sensing and Spatial Information Sciences, Volume XL-2, 2014 ISPRS Technical Commission II Symposium, Toronto (Canada), 79-86, 2014.

9) Note on line 492: Much research has been conducted and published (mostly in German or Polish language) on complex mining geometries and multi-seam mining in the Polish and German hard coal industry by e.g. H. Kratzsch or A. Sroka. Only some references are available in English language. This is a good suggestion. I have added the following reference (as part of the discussion in Sect. 5 on the various approaches worldwide): Preusse, A., Kateloe, H.J., and Sroka, A.: Subsidence and uplift prediction in German and Polish hard coal mining. Das Markscheidewesen 120, no. 1: 23-34, 2013.

Anonymous Referee #2

General comments Very interesting article, especially since the issue of the residual subsidence and the uplift of the area after mining activity in Europe is increasingly important due to the fairly widespread liquidation of active underground European mining. The paper address relevant scientific and technical questions within the scope of NHESS. The paper present new data and results. There is up to international standards. The methods and assumptions are valid and clearly outlined. The results are sufficient to support the interpretations and the conclusions. The author reaches substantial conclusions. The description of the data, the method and the results obtained is sufficiently complete and accurate to allow their reproduction by fellow scientists. The title clearly and unambiguously reflect the contents of the paper. The abstract provide a concise, complete and unambiguous summary of the work and the obtained results. The title and the abstract are pertinent, and easy to understand to a wide and diversified audience. The overall presentation is well structured, clear and easy to understand by a wide and general audience. The length of the paper is adequate. The technical

language is precise and understandable by fellow scientists. I am not English but in my opinion the English language is of good quality, fluent, simple and easy to read and understand by a wide and diversified audience. I also thank the second referee for the positive global evaluation.

Specific comments 1) It is interesting how accurate is the method of interferometry especially for just such analysis. It would be interesting, if possible, to compare the measurement results with the results of measurements of the classical levelling method. I understand the referee. It is a remark which is often made and justified. Therefore, the following text has been added at the end of section 2, referring to other published research by us: An accuracy of 1 mm/year was confirmed (Vervoort and Declercq, 2016) by comparing the INSAR-data to the GPS-reference points of the Belgian National Geographic Institute (NGI), linked to the reference stations of the Flemish Positioning Service (FLEPOS). For example the reference point HQ10 of NGI, situated 20 cm above the ground level, within the same coal mine basin was compared to the three closest reflectors surrounding this reference point and situated at a maximum horizontal distance of 35 m. For a 5-year period of uplift, 31.0 mm was measured at the NGI reference point, while the INSAR-reflectors showed a movement of respectively 25.5 mm, 29.1 mm and 29.9 mm. As the location of the reference point and the three reflectors are different (plus different size between reference point vs. reflectors), it is normal that the four values are different, but the difference is small, showing that the INSAR-data are precise for the purpose of this research.

2) Conclusions are interesting but also intuitive. If uplift is associated with swelling of clay minerals that should be linked to the phenomenon of rising water levels and the occurrence of these minerals in the geological layers. As noted in an article in Carboniferous there are no clay layers. These occur in the overburden. Therefore, uplift should be associated with the liquidation of depression cone throughout its previous range, bevor mining activity has been finished. Uplift should occur actually in the area where there is no mined out coal seams - that is just around the shaft because there

is still a safety pillar of the vertical shaft. The caving above exploited coal seams serve as ways of spread of water. Therefore, one should not expect the uplift especially above areas of former mining operation. Subsidence and uplift there are independent phenomena, only slightly linked by the mining operation. This is somewhat due to the paper, and was confirmed by the Author of the article. I fully agree with the last two sentences: the referee confirms one of the main findings of the research, i.e. there is no direct link between residual subsidence and uplift. Referring to the remark about the clay layers, there is a misunderstanding. When talking about the swelling of clay minerals, I refer to the swelling of clay minerals in argillaceous rocks, like siltstone and shale. These are present in the coal strata. So, I am not referring to clay layers in the overburden. I have checked the entire paper and I have changed the wording in the Abstract, the Introduction and the Conclusions, as the old formulation could have lead to confusion: the clay minerals in the argillaceous rocks in the coal strata I have also changed slightly the geological description of the coal strata: The waste rock within these coal strata is composed mainly of argillaceous rocks, like shale and siltstone, and of sandstone and thin (unmined) coal layers. I don't want to go as far in formulating conclusions as the referee does, by stating that the shaft pillar is the main cause for creating large uplift values. I prefer to first follow in future research the suggestion by the first referee on the multi-variate problem (see Comment 8 of first referee and my reaction).

3) It would be interesting to analyse the uplift of the ground in respect to the rising of water levels in different aquifers. This comment is, at least partly, linked to the misunderstanding about the clay and the depression cones formed (see previous point). On the other hand, comparing the uplift with the water level in the deep underground would have been interesting. As mentioned in the paper, the water level in the deep underground is on the Belgian side of this coal basin not measured; however, it has been sufficiently observed in the Dutch studies of the same coal basin, which are mentioned and discussed in the paper (see references Bekendam and Pöttgens, 1995; Caro Cuenca et al., 2013; de Vent and Roest, 2013). The suggestion by the referee,

i.e. a comparison of the surface movement with a change in water level in the aquifers would of course be interesting, but this is not directly the topic of this research,

4) The phenomenon of the residual subsidence is time dependent, it is obvious and has been stressed in the article. Therefore, it is difficult to assess real residual subsidence above the area of different mine panels, each of which ended its activity at different periods of time. On the contrary, the assessment of uplift is associated with a rise of the water and it can be well assessed after 1992 when pumping of the mine water has been finished. I agree with the referee, but I assume the referee is not suggesting here that something must be changed or added.

5) It is not easy to understand the sequence of Figures 3a and 3b, and 6a and 6b. In Fig. 3a and 3b the phenomena are presented chronologically and 6a/6b contrariwise, it is very difficult to interpret. Perhaps it makes sense but I have not found a justification for such order. This is in fact the same remark as Comment 2 by the first referee, although it is formulated differently. I assume that the changes made following Comment 2 by the first referee (see above) gives the necessary justification for the order. In addition and to make sure that the readers will understand it correctly, I have added some sentences in the discussion of Figs. 2-3, respectively Figs. 5-6: In sect. 3.1 It also justifies the choice of first considering the first 5 years instead of the entire observation period. and As mentioned earlier, all scales were halved to make the comparison easier and the main reason for considering two time zones was that we expected already a significant number of reflectors with uplift at the end of the first observation period. In sect. 3.2 Only six of the 1,808 reflectors had a slight downwards movement over this time period. This justifies the choice of first looking at the last 5 years of the second observation period. and There was a clear difference between the start and end of the second observation period, justifying the splitting of the entire observation period in two.

6) In the list of references there is no one position of Polish literature. Knothe theory, which was crucial for the prediction of subsidence has been recalled from Chinese

literature, what is curious (587-589). What the referee calls Chinese literature is a paper published in Int. J. of Rock Mech. Min. Sc in 2015. In the latter the authors refer to a publication of 1953 by Knothe. As I do not expand the theory by Knothe, I do not think it is necessary to include the original reference by Knothe,

Technical correction 1) Figures 3 and 6 are difficult to read. Probably the coloured symbols would be better. And perhaps Author should use some other symbols to display the different settlement and uplift classes? I can be wrong, but I assume that the referee has looked at a black & white print of the paper. As it is an electronic journal, I assume that the large majority of the readers will look at the digital format, which is in color. So, I have not made any changes, but of course if the editor would like me to make changes, please let me know.

2) To assess whether there is a relationship between the operation and the residual subsidence and uplift in Figures 3 and 6 the contours of operation should be shown. Without this the assessment is very difficult; One always has to try to find a compromise between presenting as much information as possible on a graph and keeping everything visible and clear. The various maps (Figs. 1, 3, 6 and 12) are all plotted on the same scale, with the same axes and the same gridlines; this has be done, just to make it possible to compare the various maps. By integrating Fig. 1 into Figs. 3 and 6, one does not improve or facilitate the assessment of possible relations between the mining characteristics and the surface movement, as the mining characteristics cannot be easily plotted on a 2D map. And also just the contours of operation is only one aspect of the mining (others are number of seams, total mining height, depth interval, time interval, etc.). To assess the various relations, I have opted to present the information in Table format for various locations and study the relationship on this basis. Again what referee 1 suggested (see comment 8 above) on the multi-variate problem could indeed improve the assessment, but I consider this as future research (and I assume referee 1 too).

3) On the Figures 1, 3 and 6 there is no section line drawn. It is a pity, because it

would make easier the interpretation of the results presented on Figures 4, 7 and 8; This could be done, but I would it then only do on Figure 1, plus I would repeat this Figure (i.e. a new Figure before the old Figure 4) and not overloading the original Fig.1 with various lines; similar to Fig. 12, which is Fig. 1 plus the position and labels of the various locations considered in the analysis of relationships. This new Figure would look like the following, which so far I have not yet integrated in the revised paper: [See attached] I am rather in favor of not including this new Figure, but I leave it up to the editor to decide on it.

Please also note the supplement to this comment:
http://www.nat-hazards-earth-syst-sci-discuss.net/nhess-2016-134/nhess-2016-134-AC1-supplement.pdf

[Figure]

**Fig. 1.** possible new figure

**Supplement:**

**nhess-2016-134 Author comments on Referees comments for paper "Surface movement above an underground coal longwall mine after closure", A. Vervoort**

**Dear Editor**

First of all, I would like to thank both referees for their comments and suggestions (no other comments were posted by the scientific community). Overall, I consider their reviews as positive. Below I write my reaction, point per point, on their points of criticism and on their suggestions. Mostly, I agree with them and I have made already the necessary changes in the manuscript (see file attached), although this was not required yet.

I have used the following color code: the text by referees is in black; my comments are in blue and the changes made in the manuscript are in red.

I hope that you agree with the changes that I have made. However, if you would still like further improvements, please let me know.

Kind regards, André Vervoort

**Anonymous Referee #1**

Received and published: 2 June 2016

The presented topic is of high significance. It addresses environmental impacts of underground coal mining after closure in terms of residual subsidence and uplift and contributes as such to after mine care and safety. The paper is well presented and structured, includes a thorough analysis of spatial data leading to new insights into mechanisms controlling vertical ground movements after mine closure.

I thank the first referee for the positive global evaluation.

From the reviewer's point of view, view minor changes may support the contribution: 1) It is advised to use passive voice consistently in the text, no use of we I know that this point is often a point of discussion and that some journals prefer the passive form and others the active one. I have checked some papers in nhess and I see both forms being published. So, I have made in the revised version no changes to the manuscript on this point. However, if the editor would insist on the passive form, please let me know and I will the necessary changes.

2) Lines 143 – 146 appear to the reader a bit unclear (overlap in time zones) – this could be rewritten more clearly

I accept this suggestion and I have reformulated this part, so that it is better formulated. The new text is as follows:

In a similar way, it was observed that, at the start of the second period of observation (from December 2003 through October 2010), certain reflectors were still undergoing downward movement. Therefore, in first instance, we looked at 5- year time zones in each observation period, which can be considered to be characterized by a pure downward movement (for the first observation period from mid-August 1992 through mid-August 1997) or a pure upwards movement (for the second observation period from mid-September 2005 through mid-September 2010). The remaining part of each observation period was also studied and for comparison purposes, a length of 2.5 years was chosen, i.e. the last 2.5 years of the first period and the first 2.5 years of the second period. These two 2.5-year time zones were from July 1998 through December 2000 and from December 2003 through June 2006, respectively. As the total first observation period was longer than 7.5 years and the second shorter, there was a gap between the time zones of 5 and 2.5 years in the first period and a small overlap in the second period, but the main advantage of doing so was that all time zones could be compared more easily.

3) At the beginning of section 3 the reader may already ask himself about the sequence of mining as it may likely influence the timely subsidence/uplift behaviour. Although discussed later in section 4, it may help to introduce basic mining parameters (depth, thickness, and mining sequence) already in the introduction section next to the description of geology. A map containing years of production of the panels may help visualising a possible link between ground movement and extraction sequence.

I have tried to build the paper in a systematic order, whereby the analysis of the data on the surface movement is the central and most important part, followed by an interpretation of the observations. On the other hand, the basic mining parameters were already mentioned in the introduction:

At a certain X-Y position within the mined area, one to eight different coals seams were mined. The combined mining height of the several seams varied from 2.0 to 12.3 m within this area. The height of the mining of individual panels varied from 0.9 to 2.7 m, and, normally, about 10 to 40 cm of it were layers of waste rock. In some cases, either no waste rock was mined or only a few centimeters were mined, but, in other cases, almost 1 m of waste rock was recorded as having been mined. As the map indicates, certain zones were not mined. Apart from the zone around the vertical shafts (around the coordinates of Latitude 51.025°N and Longitude 5.370°E), these unmined zones mainly were areas around faults. The latter were composed of a predominant set of NNW - SSE striking normal faults with subordinate N-S to NE-SW striking thrust faults. In the later decades of production, a typical longwall panel had dimensions of 200 by 800 to 1,000 m. The main and tail gates were immediately adjacent to the panel, and they were just single tunnels with a horseshoe cross-section. So, no barrier or remnant pillars existed between the

longwall panels. In the area that we studied, the mining depth varied from 539 to 967 m, and the mining occurred between 1932 and 1992. However, most of the panels were mined in the 1960s and 1970s. In Sect. 4, more details of the mining characteristics are provided, and their possible influences on the surface movements are discussed.

So, I would like to keep it this way.

Regarding the remark on adding the year of production to each individual panel on Fig. 1: I understand the referee, but I have tried this before but this leads to, on the one hand, an overloaded map with too much information and, on the other hand, not all information can be made visible. The main reason is that up to 8 panels are mined above each other. In fact one needs to see it in 3D. So, the presentation of the information in Tables (see Table 3 and 4) seems me to be the best option, although it is less visual.

4) Sentence in line 185 related to EW section could be left out (Ideally, : : :), since no section is presented.

The word 'section' has been replaced and the sentence starting with 'Ideally' has been taken out. The new text is as follows:

To better visualize the map of reflectors, the movement along a north-south line is presented in Fig. 4a. To have a sufficient amount of reflectors along this line, a north-south zone was selected for Longitude between 5.37°E and 5.38°E, about 700 m wide in the east-west direction.

5) Formulation in line 263 appears unfortunate: Better: 'It becomes obvious that an uplift over the entire area took place'

I assume the referee refers to line 213. I have changed the sentence as suggested.

In the five-year time zone from mid-September 2005 through mid-September 2010 at the end of the second observation period, it becomes obvious that an uplift over the entire area took place (Fig. 5a).

6) General question on section 3: Does the author suspect that when using different time periods (e.g. 2010 - 2013 or before 1992) results may differ, especially with respect to the rate of residual subsidence and uplift, e.g. due to the extraction sequence? In other words, is the comparison between 2 x 5 year periods just a snapshot or can observations be generalized? A short discussion may address this issue (partly later answered in section 4.3). This is an interesting and important point that the referee raises. The period of 5 years is not

chosen at random. It is based on earlier research (see Vervoort and Declercq, 2016), in which annual increases in movement were studied in detail. This showed that during both periods of about 5 years there was either only subsidence (first observation period), or only uplift (second observation period). So, that means that if we would have chosen 4 years instead of 5 years, we would have also a pure downward or upward movement, but this would be not the case by

choosing 6 or 7 year time zones. In the latter cases, the movement would change direction during the entire period (at least at a significant amount of reflectors).

The first sentence of section 3 has been changed and the changes made following comment 2 (see above) should also help to clarify this point.

Earlier research (Vervoort and Declercq, 2016) looked at annual increases in surface movement.

For your information, the paper Vervoort and Declercq, 2016 is in the mean time accepted for publication. This has also been indicated in the list.

Vervoort, A., and Declercq, P.-Y.: Surface movement above old coal longwalls after mine closure. Accepted (May 6th 2016) for publication in Int. J. of Min. Sc. and Techn., 2016.

7) Interpretation lines 356-359: Note that especially in the centre of the field, considered the x and y coordinates for min and extensive mining activities are rather close. From the map the distance appears to be less than the area of influence. Movements may be superimposed and the ground movement behaviour at a min location is most likely affected by the subsidence from a neighbouring panel. Thus the ground movements of points which are close to each other may not interpreted independently. Conclusions drawn can be associated with some fuzziness, which would reflect the limit of this study by not taking the spatial nature (i.e. angel of draw of 45 degrees and superimposition of effect of multiple panels) into account.

I fully agree with the comments by the referee. If one would consider the zone of influence of all panels, nearly all points within the area studied are within the zone of influence of at least one panel, of course sometimes a very old panel. An angle of draw of  $45^{\circ}$  means that the extent of the zone of influence beyond the edge of the panel is the depth of the panel. By considering the maximum depth at a certain location, the zones of influence are about 700 to nearly 1,000 m wider than the panels (e.g. on Fig. 1).

On the other hand, theory tells us that the maximum subsidence is of course larger above the mined out area than at the edges of the zone of influence. In one way, this is also logic. However, by looking at residual subsidence this cannot be clearly concluded, which is one the main findings of this research.

To address the point raised by the referee, a sentence has been added in Sect. 4.2:

By looking at the map of the exploitation (Fig. 12), this means that nearly the entire area studied is within the zone of influence of at least one longwall panel.

Also, earlier (section 4.1), I have added the following sentences:

Table 3 provides a summary of some basic information on the exploitation just underneath the two locations. The surface movement is of course not only affected by the mining directly below, but also by the mining around the locations. For an angle of draw of 45° the extent of the zone of influence is even equal to the depth of mining. However, the impact decreases when moving away from a panel, which justifies considering the exploitation in the immediate vicinity.

This point is further addressed under comment 8, below. It would also be worthwhile to conduct in future research a similar investigation as the one presented, but whereby data are public on the initial subsidence too, so that a comparison can be made between initial subsidence, residual subsidence, uplift and mining parameters.

8) Conclusions: One main conclusion is that there is obviously no one-to-one relation between some parameters and the effects of residual and uplift. A suggestion may be added to further research on this multi-variate problem using techniques including unsupervised learning and supervised learning.

This is a very good suggestion and the following sentence and reference has been added in the Conclusions:

As for this area, no one-on-one relationships could be clearly indentified between the surface movement and the mining characteristics, future research of this multi-variate problem could benefit by using techniques including unsupervised learning and supervised learning (Noack et al., 2014). Best would be to have data on the initial subsidence, the residual subsidence and the uplift, combined with data on the mining characteristics.

Noack, S., Knobloch, A., Etzold, S.H., Barth, A., and Kallmeier, E.: Spatial predictive mapping using artificial neural networks. In: Proceedings of the International Archives of the Photogrammetry, Remote Sensing and Spatial Information Sciences, Volume XL-2, 2014 ISPRS Technical Commission II Symposium, Toronto (Canada), 79-86, 2014.

9) Note on line 492: Much research has been conducted and published (mostly in German or Polish language) on complex mining geometries and multi-seam mining in the Polish and German hard coal industry by e.g. H. Kratzsch or A. Sroka. Only some references are available in English language.

This is a good suggestion. I have added the following reference (as part of the discussion in Sect. 5 on the various approaches worldwide):

Preusse, A., Kateloe, H.J., and Sroka, A.: Subsidence and uplift prediction in German and Polish hard coal mining. Das Markscheidewesen 120, no. 1: 23-34, 2013.

**Anonymous Referee #2**

Received and published: 15 June 2016

General comments Very interesting article, especially since the issue of the residual subsidence and the uplift of the area after mining activity in Europe is increasingly important due to the fairly widespread liquidation of active underground European mining. The paper address relevant scientific and technical questions within the scope of NHESS. The paper present new data and results. There is up to international standards. The methods and assumptions are valid and clearly outlined. The results are sufficient to support the interpretations and the conclusions. The author reaches substantial conclusions. The description of the data, the method and the results obtained is sufficiently complete and accurate to allow their reproduction by fellow scientists.

The title clearly and unambiguously reflect the contents of the paper. The abstract provide a concise, complete and unambiguous summary of the work and the obtained results. The title and the abstract are pertinent, and easy to understand to a wide and diversified audience. The overall presentation is well structured, clear and easy to understand by a wide and general audience. The length of the paper is adequate. The technical language is precise and understandable by fellow scientists. I am not English but in my opinion the English language is of good quality, fluent, simple and easy to read and understand by a wide and diversified audience. I also thank the second referee for the positive global evaluation.

**Specific comments**

1) It is interesting how accurate is the method of interferometry especially for just such analysis. It would be interesting, if possible, to compare the measurement results with the results of measurements of the classical levelling method.

I understand the referee. It is a remark which is often made and justified. Therefore, the following text has been added at the end of section 2, referring to other published research by us: An accuracy of 1 mm/year was confirmed (Vervoort and Declercq, 2016) by comparing the INSAR-data to the GPS-reference points of the Belgian National Geographic Institute (NGI), linked to the reference stations of the Flemish Positioning Service (FLEPOS). For example the reference point HQ10 of NGI, situated 20 cm above the ground level, within the same coal mine basin was compared to the three closest reflectors surrounding this reference point and situated at a maximum horizontal distance of 35 m. For a 5-year period of uplift, 31.0 mm was measured at the NGI reference point, while the INSAR-reflectors showed a movement of respectively 25.5 mm, 29.1 mm and 29.9 mm. As the location of the reference point vs. reflectors), it is normal that the four values are different, but the difference is small, showing that the INSAR-data are precise for the purpose of this research.

2) Conclusions are interesting but also intuitive. If uplift is associated with swelling of clay minerals that should be linked to the phenomenon of rising water levels and the occurrence of these minerals in the geological layers. As noted in an article in Carboniferous there are no clay layers. These occur in the overburden. Therefore, uplift should be associated with the liquidation of depression cone throughout its previous range, bevor mining activity has been finished. Uplift should occur actually in the area where there is no mined out coal seams - that is just around the

shaft because there is still a safety pillar of the vertical shaft. The caving above exploited coal seams serve as ways of spread of water.

Therefore, one should not expect the uplift especially above areas of former mining operation. Subsidence and uplift there are independent phenomena, only slightly linked by the mining operation. This is somewhat due to the paper, and was confirmed by the Author of the article. I fully agree with the last two sentences: the referee confirms one of the main findings of the research, i.e. there is no direct link between residual subsidence and uplift.

Referring to the remark about the clay layers, there is a misunderstanding. When talking about the swelling of clay minerals, I refer to the swelling of clay minerals in argillaceous rocks, like siltstone and shale. These are present in the coal strata. So, I am not referring to clay layers in the overburden. I have checked the entire paper and I have changed the wording in the Abstract, the Introduction and the Conclusions, as the old formulation could have lead to confusion:

the clay minerals in the argillaceous rocks in the coal strata

I have also changed slightly the geological description of the coal strata:

The waste rock within these coal strata is composed mainly of argillaceous rocks, like shale and siltstone, and of sandstone and thin (unmined) coal layers.

I don't want to go as far in formulating conclusions as the referee does, by stating that the shaft pillar is the main cause for creating large uplift values. I prefer to first follow in future research the suggestion by the first referee on the multi-variate problem (see Comment 8 of first referee and my reaction).

3) It would be interesting to analyse the uplift of the ground in respect to the rising of water levels in different aquifers.

This comment is, at least partly, linked to the misunderstanding about the clay and the depression cones formed (see previous point). On the other hand, comparing the uplift with the water level in the deep underground would have been interesting. As mentioned in the paper, the water level in the deep underground is on the Belgian side of this coal basin not measured; however, it has been sufficiently observed in the Dutch studies of the same coal basin, which are mentioned and discussed in the paper (see references Bekendam and Pöttgens, 1995; Caro Cuenca et al., 2013; de Vent and Roest, 2013).

The suggestion by the referee, i.e. a comparison of the surface movement with a change in water level in the aquifers would of course be interesting, but this is not directly the topic of this research,

4) The phenomenon of the residual subsidence is time dependent, it is obvious and has been stressed in the article. Therefore, it is difficult to assess real residual subsidence above the area of different mine panels, each of which ended its activity at different periods of time. On the contrary, the assessment of uplift is associated with a rise of the water and it can be well assessed after 1992 when pumping of the mine water has been finished.

I agree with the referee, but I assume the referee is not suggesting here that something must be changed or added.

5) It is not easy to understand the sequence of Figures 3a and 3b, and 6a and 6b. In Fig. 3a and 3b the phenomena are presented chronologically and 6a/6b contrariwise, it is very difficult to interpret. Perhaps it makes sense but I have not found a justification for such order. This is in fact the same remark as Comment 2 by the first referee, although it is formulated differently. I assume that the changes made following Comment 2 by the first referee (see above) gives the necessary justification for the order. In addition and to make sure that the readers will understand it correctly, I have added some sentences in the discussion of Figs. 2-3, respectively Figs. 5-6:

**In sect. 3.1**

It also justifies the choice of first considering the first 5 years instead of the entire observation period.

and

As mentioned earlier, all scales were halved to make the comparison easier and the main reason for considering two time zones was that we expected already a significant number of reflectors with uplift at the end of the first observation period.

**In sect. 3.2**

Only six of the 1,808 reflectors had a slight downwards movement over this time period. This justifies the choice of first looking at the last 5 years of the second observation period.

and

There was a clear difference between the start and end of the second observation period, justifying the splitting of the entire observation period in two.

6) In the list of references there is no one position of Polish literature. Knothe theory, which was crucial for the prediction of subsidence has been recalled from Chinese literature, what is curious (587-589).

What the referee calls Chinese literature is a paper published in Int. J. of Rock Mech. Min. Sc in 2015. In the latter the authors refer to a publication of 1953 by Knothe. As I do not expand the theory by Knothe, I do not think it is necessary to include the original reference by Knothe,

**Technical correction**

1) Figures 3 and 6 are difficult to read. Probably the coloured symbols would be better. And perhaps Author should use some other symbols to display the different settlement and uplift classes?

I can be wrong, but I assume that the referee has looked at a black & white print of the paper. As it is an electronic journal, I assume that the large majority of the readers will look at the digital format, which is in color. So, I have not made any changes, but of course if the editor would like me to make changes, please let me know.

2) To assess whether there is a relationship between the operation and the residual subsidence and uplift in Figures 3 and 6 the contours of operation should be shown. Without this the assessment is very difficult;

One always has to try to find a compromise between presenting as much information as possible on a graph and keeping everything visible and clear. The various maps (Figs. 1, 3, 6 and 12) are all plotted on the same scale, with the same axes and the same gridlines; this has be done, just to make it possible to compare the various maps.

By integrating Fig. 1 into Figs. 3 and 6, one does not improve or facilitate the assessment of possible relations between the mining characteristics and the surface movement, as the mining characteristics cannot be easily plotted on a 2D map. And also just the contours of operation is only one aspect of the mining (others are number of seams, total mining height, depth interval, time interval, etc.). To assess the various relations, I have opted to present the information in Table format for various locations and study the relationship on this basis. Again what referee 1 suggested (see comment 8 above) on the multi-variate problem could indeed improve the assessment, but I consider this as future research (and I assume referee 1 too).

3) On the Figures 1, 3 and 6 there is no section line drawn. It is a pity, because it would make easier the interpretation of the results presented on Figures 4, 7 and 8;

This could be done, but I would it then only do on Figure 1, plus I would repeat this Figure (i.e. a new Figure before the old Figure 4) and not overloading the original Fig.1 with various lines; similar to Fig. 12, which is Fig. 1 plus the position and labels of the various locations considered in the analysis of relationships.

This new Figure would look like the following, which so far I have not yet integrated in the revised paper:

I am rather in favor of not including this new Figure, but I leave it up to the editor to decide on it.

[revised manuscript text omitted]
. An accuracy of 1 mm/year was 128 129 confirmed (Vervoort and Declercq, 2016) by comparing the INSAR-data to the GPS-reference points of the Belgian National Geographic Institute (NGI), linked to the reference stations of the 130 Flemish Positioning Service (FLEPOS). For example the reference point HQ10 of NGI, situated 131 20 cm above the ground level, within the same coal mine basin was compared to the three closest 132 reflectors surrounding this reference point and situated at a maximum horizontal distance of 35 133 m. For a 5-year period of uplift, 31.0 mm was measured at the NGI reference point, while the 134 INSAR-reflectors showed a movement of respectively 25.5 mm, 29.1 mm and 29.9 mm. As the 135 location of the reference point and the three reflectors are different (plus different size between 136 reference point vs. reflectors), it is normal that the four values are different, but the difference is 137 small, showing that the INSAR-data are precise for the purpose of this research. 138

- 139
- 140

**141 **3 Analysis of surface movement**

142

Earlier research (Vervoort and Declercq, 2016) looked at annual increases in surface movement. 143 144 It showed that, in this area at the end of the first period of observation (from August 1992 145 through December 2000), uplift had already been initiated in certain zones or for certain reflectors. In a similar way, it was observed that, at the start of the second period of observation 146 147 (from December 2003 through October 2010), certain reflectors were still undergoing downward movement. Therefore, in first instance, we looked at 5- year time zones in each observation 148 period, which can be considered to be characterized by a pure downward movement (for the first 149 observation period from mid-August 1992 through mid-August 1997) or a pure upwards 150 movement (for the second observation period from mid-September 2005 through mid-September 151 2010). The remaining part of each observation period was also studied and for comparison 152 purposes, a length of 2.5 years was chosen, i.e. the last 2.5 years of the first period and the first 153 2.5 years of the second period. These two 2.5-year time zones were from July 1998 through 154 December 2000 and from December 2003 through June 2006, respectively. As the total first 155 observation period was longer than 7.5 years and the second shorter, there was a gap between the 156 time zones of 5 and 2.5 years in the first period and a small overlap in the second period, but the 157 main advantage of doing so was that all time zones could be compared more easily. Hence, all 158 159 scales for the graphs that correspond to the 2.5-year time zones are halved.

160 In this research, downward movement has a negative sign, and uplift has a positive sign; the

same convention was used for the rate of movement (e.g., per year). However, when discussing

the smallest (minimum) movement or the largest (maximum) movement, we considered the

absolute value of the movement; in other words, when discussing the minimum rate, we did not

apply the pure mathematical definition of minimum. For the area studied, no public data were

- available for the subsidence that occurred prior to the satellite observations.
- 166

**167 **3.1** First observation period, characterized, on average, by subsidence**

168

169 In the five years from mid-August 1992 through mid-August 1997, the area studied was

170 characterized by an overall downward movement (Table 1 and Fig. 2a). Only two out of 1,073

reflectors were characterized by small upward movements, i.e., 3 and 6 mm. In the overall

172 picture, these can be neglected. It also justifies the choice of first considering the first 5 years

instead of the entire observation period. Among the reflectors, 69% underwent residual

[revised manuscript text omitted]
- but also by the mining around the locations. For an angle of draw of  $45^{\circ}$  the extent of the zone of
- 318 influence is even equal to the depth of mining. However, the impact decreases when moving
- away from a panel, which justifies considering the exploitation in the immediate vicinity. Fig. 12

indicates both locations. Under the maximum of the residual subsidence, the mining was more 320 recent than under the maximum of the uplift. Mining took place in the periods of 1968-1982 and 321 1939-1959, respectively. However, 1982 was still 10 years before closure (and the start of 322 observation). A corner of a panel, which was mined in 1992 at a depth of 820 m, is situated at 323 324 about 250 m to the west of the location Max RES SUBS. This means that this location is within the zone of influence of that panel. However, on the E-W transect (across the panel), we did not 325 observe any maximum in residual subsidence above the most recent panel. When comparing the 326 327 mining depth, mining height, and the number of panels mined underneath the two locations, the 328 mining characteristics were rather similar. So, this means that, apart from possibly the time of 329 mining, there was no clear indication concerning the causes of the difference between the 330 movements of the two locations. In the next two paragraphs, more locations are compared, which will indicate whether the effect of the time of mining is significant. 331

[revised manuscript text omitted]

**Figures**

---

## Author Comment (AC2) · 5 Jul 2016

see author comments on both referee reports

---

## Author Comment (AC3) · 5 Jul 2016

see author comments on both referee reports